# INVERSELY LEARNING TRANSFERABLE REWARDS VIA ABSTRACTED STATES

## ABSTRACT

Inverse reinforcement learning (IRL) has made significant progress in recovering reward functions from expert demonstrations. However, a key challenge remains: how to extract reward functions that generalize across related but distinct task instances. In this paper, we address this by focusing on transferable IRL—learning intrinsic rewards that can drive effective behavior in unseen but structurally aligned environments. Our method leverages a variational autoencoder (VAE) to learn an abstract representation of the state space shared across multiple source task instances. This abstracted space captures high-level features that are invariant across tasks, enabling the learning of a unified abstract reward function. The learned reward is then used to train policies in a separate, previously unseen target instance without requiring new demonstrations in the target instance. We evaluate our approach on multiple environments from Gymnasium and AssistiveGym, demonstrating that the learned abstract rewards consistently support successful policy learning in novel task settings.

## 1 INTRODUCTION

The objective of inverse reinforcement learning (IRL) is one of abductive reasoning: to infer the reward function that best explains the observed trajectories. This is challenging because the available data is often sparse, which admits many potential solutions (some degenerate), and the learned reward functions may not generalize for use in target instances that could be slightly different. Despite these challenges, significant progress has been made in the last decade toward learning the underlying reward functions in both discrete and continuous domains. A key advance in IRL next is to learn reward functions that represent *intrinsic preferences*, which become relevant in aligned task instances not seen previously. This contributes to the transferability of the learned rewards – an important characteristic of a general solution.

In this paper, we introduce a new method that generalizes IRL to previously unseen tasks but which exhibit commonality with the observed ones in terms of shared intrinsic (or core) preferences. Abstractions offer a powerful representation toward generalization (Allen et al., 2021), and so we introduce the novel concept of an *abstract reward function*. To illustrate, consider the Ant domain from OpenAI Gymnasium (Schulman et al., 2016) and two Ant environments with differing pairs of disabled legs as the source environments and an Ant environment with another pair of disabled legs as the target. As the source and target ants have different disabled legs, the marginal state distributions of the sources are different from the target's, which makes it difficult to transfer a learned reward function. However, if we focus on the ant's torso instead of its legs, the marginal state distribution of the torso remains mostly the same across both the sources and the target environment. So, inversely learning a reward function based on the torso, which is the *abstraction*, allows the function to be transferred across any disabled leg. Our method utilizes observed behavior data from two or more differing task instances of a domain as input to a variational autoencoder (VAE). A single encoder is coupled with multiple decoders, one for each source instance, to reconstruct the instance trajectories. We show how the common latent variable(s) of this distinct VAE model can be interpreted and shaped as an abstract reward function that governs the input task behaviors. Note that two or more aligned task behaviors are needed to learn the shared intrinsic preferences to perform the tasks.

We evaluate our method for *transferable IRL*, labeled TraIRL, on multiple benchmarks: Gymnasium domains (Schulman et al., 2016) and the robotic AssistiveGym (Erickson et al., 2019). We utilize

trajectories from two differing instances in each domain as input to the VAE and show how the inversely learned abstract reward function can help successfully learn a high-quality behavior in a third aligned instance of the domain. These results open a new frontier for methods that may learn abstract rewards via IRL to offer a level of generalizability not previously seen in the literature.

## 2 RELATED WORK

**Extant transfer learning for IRL struggles with mismatches in environment dynamics, which limits reward transferability**. Tanwani & Billard (2013) introduce an approach to learn diverse strategies from multiple experts, focusing on shared knowledge. However, it assumes unchanged dynamics between experts, which limits its applicability to dynamic environments. I2L (Gangwani & Peng, 2020) is designed for state-only imitation learning and addresses transition dynamics mismatches by using a prioritized trajectory buffer and optimizing a lower bound on the expert's state-action visitation distribution. While empirically effective, it lacks theoretical guarantees on reward transferability and does not formally justify how the learned reward generalizes across different dynamics. Viano et al. (2024) analyzes maximum causal entropy (MCE) IRL under transition dynamics mismatch, deriving necessary and sufficient conditions for its transferability and providing a tight bound on performance degradation. It proposes a robust MCE IRL algorithm but struggles to generalize under action space shifts due to its reliance on matching state-action occupancy measures.

**Rewards learned by adversarial IRL and IL methods may not transfer across environments**. AIRL (Fu et al., 2018), $f$-MAX (Ghasemipour et al., 2020) and $f$-IRL (Ni et al., 2021) claim that their learned reward functions generalize to unseen or dynamically different environments. But, these claims are not supported by explicit structural frameworks or theoretical guarantees, leaving the transferability unpredictable. Furthermore, the learned rewards are tied to specific expert policy trajectories, preventing their use in training new policies from scratch. In contrast, IQ-Learn (Garg et al., 2021) is non-adversarial and learns soft Q-functions from expert data, which offers improved stability and efficiency. However, its reliance on action-dependent Q-functions limits generalization to state-only reward functions.

**Reward identification or identifiability may not be sufficient to learn a transferable reward function**, as it focuses only on recovery the true reward function from expert demonstrations. Cao et al. (2024) mitigates reward ambiguity using an entropy-regularized framework. It relies on multiple optimal policies under varying dynamics, but the method does not focus on transferability and is not suitable for scenarios that do not have such policies. Rolland et al. (2024) presents a reward identification approach for discrete state-action spaces, utilizing variations across environments. However, the method struggles with continuous states and prioritizes identifiability over transferability. Kim et al. (2021) formalize reward identifiability in deterministic MDPs using the maximum entropy objective, and provide conditions for identifiability. But, it does not address the challenge of creating transferable reward representations as we do in this paper through abstraction.

**Existing successor feature matching algorithms lack transferable feature functions, which limits their ability to generalize to unseen tasks.** SFM (Jain et al., 2025) introduces successor feature matching into IRL while avoiding adversarial training. However, its feature functions are not designed for transfer learning. Our method can be viewed as an extension of SFM, where we incorporate a transferable abstract feature function. In Appendix D.2, we compare SFM as a backbone with $f$-IRL as a backbone.

## 3 BACKGROUND

We briefly review MCE IRL (Ziebart et al., 2010) as it informs our method. The entropy-regularized Markov decision process (MDP) is characterized by the tuple $(\mathcal{S}, \mathcal{A}, \mathcal{T}, r, \gamma, \rho_0)$. Here, $\mathcal{S}$ and $\mathcal{A}$ denote the state and action spaces, respectively, while $\gamma \in (0, 1)$ is the discount factor. In the standard RL context, the dynamics modeled by the transition distribution $\mathcal{T}(s'|a, s)$, the initial state distribution $\rho_0(s)$, and the reward function $r(s, a)$ are unknown, and can only be determined through interaction with the environment. Optimal policy $\pi$ under the maximum entropy framework maximizes the objective $\pi^* = \arg\max_\pi \mathbb{E}_{\tau \sim \pi} \left[ \sum_{t=0}^{T} \gamma^t (r(s_t, a_t) + H(\pi(\cdot|s_t))) \right]$, where $\tau \triangleq (s_0, a_0, ..., s_T, a_T)$

denotes a sequence of states and actions induced by the policy and transition function, and $H(\pi(\cdot|s))$ is the entropy of the action distribution from policy $\pi$ for state $s$.

Another method, $f$-IRL (Ni et al., 2021), integrates $f$-divergence to improve scalability and robustness. $f$-IRL relies on a generator-discriminator schema to recover a stationary reward function by matching the expert's state marginal distribution (also called *state density* or *occupancy distribution*) – an approach that builds and improves on the state marginal matching (SMM) algorithm (Lee et al., 2019). We rely on a variant of $f$-IRL that minimizes the 1-Wasserstein distance between the state marginals, as this distance is an integral probability metric:

$$\mathcal{L}_{\mathcal{F}}(\boldsymbol{\theta}) = \mathcal{D}_{\mathcal{F}}(\rho_E || \rho_{\boldsymbol{\theta}}) = W_1(\rho_E(s), \rho_{\boldsymbol{\theta}}(s)), \tag{1}$$

where $\mathcal{D}_{\mathcal{F}}$ is a divergence measure between distributions, $W_1$ is the 1-Wasserstein distance, $\rho_E$ and $\rho_{\boldsymbol{\theta}}$ denote the state densities of the expert and the soft-optimal learner under the reward function $R_{\boldsymbol{\theta}}$. Another advantage of $f$-IRL is its separation of the reward and discriminator networks, effectively forming a distillation model. The discriminator serves as the stronger model, while the reward function distills its information, yielding better generalization than the single-discriminator design used in most adversarial IRL methods.

# 4 LEARNING TRANSFERABLE REWARDS VIA ABSTRACTION

We introduce our method for transferable IRL, labeled TraIRL, in this section. The approach learns a state-only abstracted reward function optimized for transfer from expert trajectories in source tasks. This reward function is then employed to learn a well-performing policy in the target tasks.

## 4.1 PROBLEM DEFINITION

We are provided with a set of expert trajectories induced by policies of multiple source MDPs $\{\mathcal{M}^i\}_{i=1}^n$, each corresponding to a distinct but related task. The goal is to infer a shared intrinsic reward function that also generalizes to a previously unseen target MDP $\mathcal{M}_T$, allowing an agent to perform the task effectively in a target task without access to expert demonstrations there.

To support such transfer, we define a shared *abstracted state space* $\bar{\mathcal{S}}$ that captures common, task-invariant features across the source MDPs. This is formalized below:

**Definition 1** (Cross-Task Abstraction). *Let $\mathcal{M}^i = (\mathcal{S}^i, \mathcal{A}^i, \mathcal{P}^i, \mathcal{R}^i, \gamma^i, \rho_0^i)$ be the ground MDP for task $i$. A cross-task abstraction is defined by a mapping $\phi : \mathcal{S}^i \to \bar{\mathcal{S}}$, where $\phi(s^i) \in \bar{\mathcal{S}}$ denotes the abstracted state corresponding to the ground state $s^i$. The inverse mapping $\psi^i(\bar{s})$ denotes the set of ground states in task $i$ that are mapped to the abstract state $\bar{s} \in \bar{\mathcal{S}}$.*

Our objective is to learn a state-only abstract reward function $\bar{\mathcal{R}} : \bar{\mathcal{S}} \to \mathbb{R}$ and a cross-task abstraction $\phi$ such that $\bar{\mathcal{R}} \circ \phi : \mathcal{S}^i \to \mathbb{R}$ when used to substitute $\mathcal{R}^i$ in $\mathcal{M}^i$ induces expert behaviors across source tasks $i = 1$ to $n$. The abstract reward function $\bar{\mathcal{R}}$ and the cross-task abstraction $\phi$ can be transferred to a target task where expert demonstrations are not available.

## 4.2 LEARNING ABSTRACTION VIA MULTI-HEAD VAE

To enable reward transfer across different tasks, it is essential to extract an abstract representation that captures intrinsic, task-invariant structure from expert demonstrations. In TraIRL, we implement the *cross-task abstraction function* $\phi$, introduced in Def. 1, using the *encoder* of a variational autoencoder (VAE) (Kingma & Welling, 2014). Specifically, the encoder $p_{\phi}(z|s)$ maps a ground state $s \in \mathcal{S}^i$ to a latent variable $z \in \bar{\mathcal{S}}$, which serves as the abstracted state $\phi(s)$. The corresponding *decoder*, denoted $q_{\psi}(s|z)$, approximates the inverse mapping $\psi(z)$, reconstructing the original ground state from its abstracted state.

To generalize across tasks, we employ a *single task-agnostic encoder* $p_{\phi}$, which is shared across all source tasks, and $n$ *task-specific decoders* $\{q_{\psi^i}\}_{i=1}^n$, where each decoder specializes in reconstructing states in the $i$-th source task. During training, the shared encoder learns to extract abstract states across tasks, while each decoder captures task-specific reconstruction from the shared abstraction.

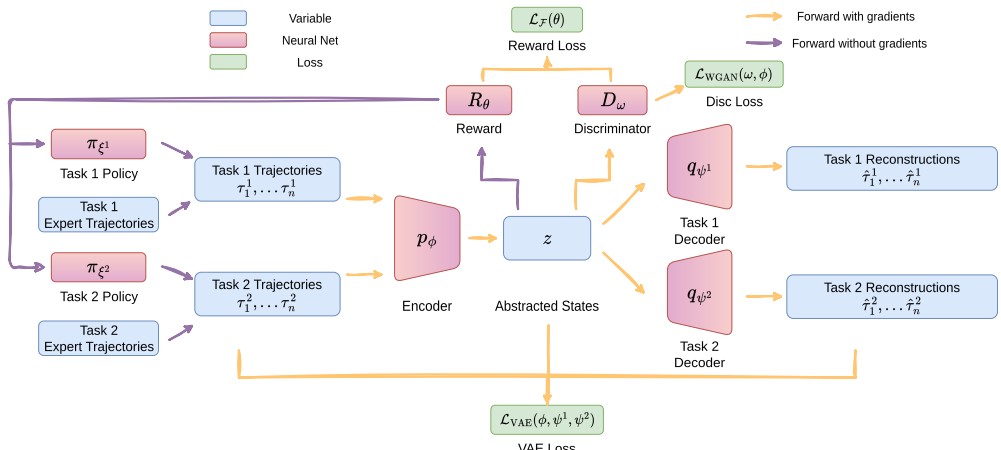

Figure 1: TraIRL framework overview. Expert and learner trajectories in multiple source tasks are mapped to shared abstract states via a shared encoder. A discriminator compares the abstracted state densities to estimate the 1-Wasserstein distance between expert and learner, which guides learning a reward function over the abstract space through a covariance-based objective. The learned reward function then optimizes the learner policy.

The VAE is trained by maximizing the evidence lower bound (ELBO), which balances reconstruction accuracy with latent space regularization. For our multi-task setting, the overall VAE objective is:

$$\mathcal{L}_{\text{VAE}}(\boldsymbol{\phi}, \boldsymbol{\psi}^1, \dots, \boldsymbol{\psi}^n; \boldsymbol{\tau}) = \sum_{i=1}^{n} \mathcal{L}_{\text{VAE}}^i(\boldsymbol{\phi}, \boldsymbol{\psi}^i; \boldsymbol{\tau}^i)$$
$$= \sum_{i=1}^{n} \left( \mathbb{E}_{z \sim p_{\boldsymbol{\phi}}(z|s^i)} \left[ \log q_{\psi^i}(s^i|z) \right] - \lambda_{\mathcal{D}} \, \mathcal{D}_{\text{KL}} \left( p_{\boldsymbol{\phi}}(z|s^i) \, \| \, p(z) \right) \right), \quad (2)$$

where $\boldsymbol{\tau} \triangleq \langle \tau^1, \dots, \tau^n \rangle$ denotes the collection of expert trajectories from $n$ source tasks, and each trajectory $\tau^i = \{s_0^i, s_1^i, \dots, s_T^i\}$ consists of a sequence of states from the $i$-th source environment. In the rest of the paper, we use $s^i$ to denote a state from the $i$-th environment. The prior $p(z)$ is a normal distribution $\mathcal{N}(0, I)$, and $\lambda_{\mathcal{D}}$ controls the weight of the KL regularization term.

### 4.3 STRUCTURING THE ABSTRACT STATE SPACE

While the VAE compresses the high-dimensional state space into a lower-dimensional abstraction, it does not incorporate information about optimality, which in IRL refers to whether a state is part of an expert trajectory. The learned abstraction focuses on reconstruction and distributional coherence but lacks structure related to optimality. To address this, we introduce a discriminator-guided mechanism that encourages the abstracted state space to capture optimality by distinguishing between expert and learner trajectories.

We use Wasserstein GAN with gradient penalty (WGAN-GP) (Gulrajani et al., 2017) to estimate the 1-Wasserstein distance between the abstracted state distributions of the expert and the learner. The objective function of WGAN-GP in our multi-task setting is:

$$\mathcal{L}_{\text{WGAN}}(\boldsymbol{\phi}, \boldsymbol{\omega}; \boldsymbol{\tau}) = \sum_{i=1}^{n} \mathcal{L}_{\text{WGAN}}^i(\boldsymbol{\phi}, \boldsymbol{\omega}; \boldsymbol{\tau}^i) = \sum_{i=1}^{n} \left( \mathbb{E}_{z \sim p_{\boldsymbol{\phi}}(z|s), s \sim \rho_L(s^i)}[D_{\boldsymbol{\omega}}(z)] \right.$$
$$\left. - \mathbb{E}_{z \sim p_{\boldsymbol{\phi}}(z|s), s \sim \rho_E(s^i)}[D_{\boldsymbol{\omega}}(z)] + \lambda_{\text{GP}} \, \mathbb{E}_{z \sim \hat{\rho}(z)} \left[ (\|\nabla_z D_{\boldsymbol{\omega}}(z)\|_2 - 1)^2 \right] \right), \quad (3)$$

where $\rho_L(s^i)$ and $\rho_E(s^i)$ denote the state densities of the learner and expert in the $i$-th source task, respectively, and $\hat{\rho}(z)$ is the distribution of points $z$ sampled uniformly along a straight line between the abstract states of experts and the abstract states of learner.

A discriminator trained to distinguish between expert and learner trajectories in the abstract space imposes optimality awareness on the learned abstract state space. This structure facilitates learning a generalizable reward function and improves its transferability to target tasks.

## 4.4 Robust Transferable Reward Learning via Abstracted States

Once the abstracted state space is learned, we can recover the reward function on top of it. It is worth noting that TraIRL is not constrained to $f$-IRL; the abstracted states can be integrated with any adversarial IRL algorithm (Appendix D.2). We choose f-IRL in this work because its separate reward and discriminator networks naturally induce a distillation process, leading to better generalization, which is crucial for transfer learning. In $f$-IRL, the discriminator maximizes the 1-Wasserstein distance between the expert and learner state densities in the ground state space, Eqn.1. The reward function is then learned by minimizing this distance with respect to its parameters $\boldsymbol{\theta}$. In TraIRL, we adapt this formulation to operate over the abstracted state space. Accordingly, the reward function is defined over the abstracted state space, i.e., $R_{\boldsymbol{\theta}}(z)$, where $z \sim p_{\boldsymbol{\phi}}(z|s)$. The reward function $R_{\boldsymbol{\theta}}(z)$ is trained to minimize the 1-Wasserstein distance between the abstracted state densities induced by the learner and the expert, thus driving the learner's occupancy measure in the abstracted space to match that of the expert. The abstracted state density is defined as $\rho(z) = \int_S p_{\boldsymbol{\phi}}(z|s)\,\rho(s)\,ds$. Thus, the objective function of the reward function is:

$$\mathcal{L}_{\mathcal{F}}(\boldsymbol{\theta}) = \sum\nolimits_{i=1}^{n} W_1(\rho_E(z^i), \rho_L(z^i))$$
$$= \sum\nolimits_{i=1}^{n} \left( \mathbb{E}_{z \sim p_{\boldsymbol{\phi}}(z|s), s \sim \rho_E(s^i)}[D_{\boldsymbol{\omega}}(z)] - \mathbb{E}_{z \sim p_{\boldsymbol{\phi}}(z|s), s \sim \rho_L(s^i)}[D_{\boldsymbol{\omega}}(z)] \right). \quad (4)$$

**Theorem 1** (Gradient of Reward Function). *The analytic gradient of our objective function $\mathcal{L}_{\mathcal{F}}(\boldsymbol{\theta})$ presented in Eq.4 w.r.t $\boldsymbol{\theta}$ can be derived as:*

$$\nabla_{\boldsymbol{\theta}} \mathcal{L}_{\mathcal{F}}(\boldsymbol{\theta}) = \sum\nolimits_{i=1}^{n} cov_{s \sim \hat{\rho}(s^i), z \sim p_{\boldsymbol{\phi}}(z|s)} \left( D_{\boldsymbol{\omega}}(z), \nabla_{\boldsymbol{\theta}} R_{\boldsymbol{\theta}}(z) \right), \quad (5)$$

*where $\hat{\rho}(s^i) = \frac{1}{2}(\rho_L(s^i) + \rho_E(s^i))$.*

We derive the gradient in Appendix B.1. Theorem 1 shows that the gradient of $\mathcal{L}_{\mathcal{F}}(\boldsymbol{\theta})$ with respect to the reward parameters is given by the covariance between the discriminator output and the reward value. Since the gradient is taken only with respect to $\boldsymbol{\theta}$, the encoder $p_{\boldsymbol{\phi}}$ remains fixed during reward optimization. This design is critical for generalization to the target task for two reasons. First, the covariance aligns the reward function with the dominant structure captured by the discriminator, which reflects differences between expert and learner behaviors in the abstracted space, while ignoring outliers or overfitted features, similar to a distillation process (Hinton et al., 2015). Second, decoupling the reward function from the encoder ensures it operates on a fixed abstracted space learned from multiple source domains, preventing overfitting to source tasks and improving its transferability to the target domain.

The overall objective function for TraIRL involving $n$ source tasks is a linear combination of the three objective functions defined previously:

$$\mathcal{L}(\boldsymbol{\theta}, \boldsymbol{\omega}, \boldsymbol{\phi}, \boldsymbol{\psi}^1, \ldots, \boldsymbol{\psi}^n) = \lambda_{\text{VAE}} \mathcal{L}_{\text{VAE}}(\boldsymbol{\phi}, \boldsymbol{\psi}^1, \ldots, \boldsymbol{\psi}^n) - \lambda_{\mathcal{F}} \mathcal{L}_{\mathcal{F}}(\boldsymbol{\theta}) + \lambda_{\text{WGAN}} \mathcal{L}_{\text{WGAN}}(\boldsymbol{\phi}, \boldsymbol{\omega}), \quad (6)$$

where $\lambda_{\text{VAE}}, \lambda_{\mathcal{F}}$ and $\lambda_{\text{WGAN}}$ are the hyperparameters for $\mathcal{L}_{\text{VAE}}, \mathcal{L}_{\mathcal{F}}, \mathcal{L}_{\text{WGAN}}$, respectively.

Fu et al. (2018) highlight that disentangling rewards from task dynamics is crucial for transfer. A state-only reward is dynamics-agnostic when the decomposability condition holds. In our approach, abstracted states are trained to be invariant to dynamics and satisfy decomposability by jointly optimizing the VAE and discriminator across multiple source tasks. This enables TraIRL to learn state-only rewards that induce dynamics-agnostic behavior, improving transferability. Appendix B.3 provides proofs and an example where decomposability fails in the ground MDP but holds in the abstract MDP.

## 4.5 Analytical Framework for Reward Transferability

We aim to formally characterize when a reward function learned using TraIRL can be expected to generalize to an unseen target task.

**Definition 2** (Reward Transferability). *Define a reward function $R_{\boldsymbol{\theta}}$ learned for states $\mathcal{S}^i$ of the source task $\mathcal{T}^i$ as **transferable** to a target task $\mathcal{T}^t$ iff for a small positive constant $\epsilon > 0$,*

$$W_1(\rho^*_{\mathcal{T}^t}(z), \rho_{\mathcal{T}^i}(z)) \le \epsilon, \quad (7)$$

*where $\rho^*_{\mathcal{T}^t}(z)$ is the abstract state density induced by the (soft-)optimal policy $\pi^*_{\mathcal{T}^t}$ in the target task and $\rho_{\mathcal{T}^i}(z)$ is the abstract state density induced by the policy $\pi_{\mathcal{T}^i}$ obtained by optimizing for $R_\theta$ in the source task $\mathcal{T}^i$.*

Definition 2 states that the reward function $R_\theta$ is transferable if the policy obtained by optimizing $R_\theta$ in the source task yields an abstract state density that closely matches the one induced by the (soft-)optimal policy in the target task. Next, Theorem 2 delineates the conditions under which the reward function learned using TraIRL is transferable from source tasks to a target task.

**Theorem 2** (Applicability of TraIRL)**.** *Let $R_\theta$ denote the reward function learned by optimizing Eq. 4, $\rho^*_{\mathcal{T}^i}(z)$ denote the abstract state density of the expert in the $i$-th source task $\mathcal{T}^i$, $\epsilon$ be the positive threshold from Def. 2, and $\alpha \in (0, 1)$. If, for every $i$,*

$$W_1(\rho^*_{\mathcal{T}^t}(z), \rho^*_{\mathcal{T}^i}(z)) \leq \alpha\epsilon \tag{8}$$

$$W_1(\rho_{\mathcal{T}^i}(z), \rho^*_{\mathcal{T}^i}(z)) \leq (1-\alpha)\epsilon \tag{9}$$

*then the reward function $R_\theta$ is **transferable** to the target task, enabling effective policy learning.*

*Sketch of Proof.* Wasserstein distance, $W_1$, satisfies the *triangle inequality*. Applying it to Eqs. 8 and 9, we derive Eq. 7. This satisfies the transferable reward condition in Def 2. $\square$

Theorem 2 highlights two conditions to ensure the transferability of rewards under TraIRL. First, Eq. 8 captures the generalizability of the abstract state space (structural alignment assumption), i.e., whether optimality in the source and target tasks induce similar distributions over the shared abstract space. Second, Eq. 9 addresses the correctness of the abstraction, i.e., the optimal policy using the learned rewards in a source task induces an abstract state density that closely matches that induced by the expert policy.

## 5 EXPERIMENTS

We implement Algorithms 1 and 2 in PyTorch, and empirically evaluate the performance of TraIRL across MuJoCo benchmark domains (Todorov et al., 2012) and across the human-robot Assistive Gym domains (Erickson et al., 2019). TraIRL is run on 50 trajectories of two distinct source tasks with different dynamics, from these domains. It is run until convergence on each task. To evaluate TraIRL's reward transferability, we use forward RL in the target task with the inversely learned rewards to obtain the policy. Note that we do not have expert trajectories in the target task and we do not use the target's true rewards and utilize the learned reward function only. We measure how well the policy performs by simulating it for 10 episodes and reporting the mean and variance of the accumulated rewards.

We adopt a similar procedure for three state-of-the-art baseline techniques: AIRL-ME (Buening et al., 2022), RIME (Chae et al., 2022), and I2L. AIRL-ME is an extension from AIRL to adapt to various environments. RIME and I2L are designed to handle dynamics changes specifically and potentially offers a better generalization across tasks, though without any abstraction mechanisms, as used by TraIRL. The input to all algorithms is the default observation returned by the Gymnasium environment, including velocity, joint values, angular velocity of joints and etc.

**Model architecture and implementation** We employ a multilayer perceptron with Tanh as the activation function for both the encoder and decoder in VAE and to represent the reward function. TraIRL and all baselines have been implemented in PyTorch. We choose reverse KL divergence as the objective function in $f$-IRL because it has been shown to be robust and faster to converge for IRL compared to regular KL divergence (Ni et al., 2021; Ghasemipour et al., 2020). We use Soft Actor-Critic (SAC) (Haarnoja et al., 2018) as the backbone RL algorithm. Further details regarding the model architecture and hyperparameters to aid reproducibility are available in Appendix C.

### 5.1 EVALUATIONS IN MUJOCO-GYM

Two source tasks and one target task from each of the **Half Cheetah** and **Ant** domains in MuJoCo-Gym are used. Figure 2 illustrates the source and target tasks, which differ in dynamics between the sources and between the sources and the target. Specifically, these differences in dynamics arise from *disabling* different pairs of legs. Disabled legs are indicated in red in the frames of Fig. 2. While the

action space remains unchanged across the environments, as the input actions are directly applied to all joints regardless of whether a leg is disabled, the dynamics differ because the disabled legs cannot respond to the input actions.

Although the original reward functions across source and target tasks are similar in structure, IRL focuses on recovering a reward that matches the expert's state density. Since each task exhibits distinct dynamics, the resulting expert state densities differ. Consequently, learning reward functions independently in each task yields task-specific rewards that are not directly transferable. The experiments support this claim are reported in the Appendix D.4.

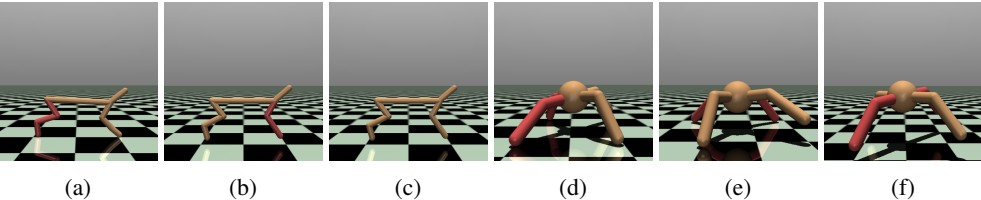

|   (a)   |   (b)   |   (c)   |   (d)   |   (e)   |   (f)   |

Figure 2: Source and target tasks from MuJoCo-Gym domains. Red legs are the disabled legs of the robots. Frames (a,b) depict source tasks of running with a disabled leg in **Half Cheetah** while (c) represents the target environment with no disability. Similarly, frames (d,e) show the source tasks of running with different pairs of disabled legs in **Ant**, whereas (f) shows the target of running with another pair of disabled legs.

### 5.1.1 REWARD TRANSFERABILITY

Tables 1 and 2 report our results using TraIRL and the baselines on Half Cheetah and Ant domains, respectively. These show the mean cumulative rewards obtained in the target environment as well as in the two source environments. The optimal policy's (expert) rewards for each are reported as well.

Table 1: Mean cumulative rewards with standard deviation for the **Half Cheetah** domain.

|  | **Sources** | | **Target** |
|---|---|---|---|
|  | Run (rear disabled) | Run (front disabled) | Run (no disability) |
| AIRL-ME | $4{,}014.52 \pm 79.3$ | $3{,}905.38 \pm 73.1$ | $3{,}725.63 \pm 80.9$ |
| RIME | $4{,}271.67 \pm 36.1$ | $4{,}129.19 \pm 83.4$ | $4{,}061.22 \pm 58.6$ |
| I2L | $4{,}396.43 \pm 45.4$ | $\mathbf{4{,}518.66 \pm 52.2}$ | $4{,}512.71 \pm 66.4$ |
| **TraIRL** | $\mathbf{4{,}404.07 \pm 57.6}$ | $4{,}359.35 \pm 99.2$ | $\mathbf{5{,}835.11 \pm 74.0}$ |
| Expert | $5{,}052.25 \pm 25.4$ | $5{,}499.07 \pm 156.1$ | $6{,}420.38 \pm 37.9$ |

Table 2: Mean cumulative rewards with standard deviation for the **Ant** domain.

|  | **Sources** | | **Targets** | | |
|---|---|---|---|---|---|
|  | Leg 1,2 disabled | Leg 0,3 disabled | Leg 1,3 disabled | Leg 0,2 disabled | Half Cheetah |
| AIRL-ME | $2{,}389.65 \pm 52.0$ | $2{,}231.09 \pm 87.6$ | $2{,}098.11 \pm 99.3$ | $2{,}190.12 \pm 50.2$ | - |
| RIME | $2{,}681.67 \pm 49.9$ | $2{,}708.14 \pm 81.5$ | $2{,}190.71 \pm 61.9$ | $2{,}188.00 \pm 59.8$ | - |
| I2L | $\mathbf{2{,}831.28 \pm 36.4}$ | $2{,}786.90 \pm 79.4$ | $2{,}585.32 \pm 84.5$ | $2{,}618.32 \pm 92.4$ | - |
| **TraIRL** | $2{,}714.18 \pm 35.9$ | $\mathbf{2{,}936.52 \pm 95.5}$ | $\mathbf{2{,}917.92 \pm 79.3}$ | $\mathbf{3{,}156.54 \pm 63.1}$ | $\mathbf{5{,}378.78 \pm 61.7}$ |
| Expert | $3{,}312.12 \pm 304.3$ | $3{,}303.99 \pm 341.0$ | $3{,}369.05 \pm 216.8$ | $3{,}590.57 \pm 158.2$ | $6{,}420.38 \pm 37.9$ |

Observe that the rewards learned by TraIRL achieve the highest average return compared to all baselines in the **target tasks** for both domains. As such, the abstracted rewards learned by TraIRL from the two sources are most transferable and correct. I2L achieves the next best performance on the target task but remain significantly lower than TraIRL's (Student's paired t-test, $p < 0.01$). More detailed experiments, including 10 source and 5 target tasks, are provided in the Appendix D.10. Please refer to Appendix D.9 for the failure case of TraIRL where the structural alignment assumption (Eq. 8) is violated.

We ambitiously evaluate TraIRL in a more challenging cross-domain setting: transfer from **Ant** to the **Half Cheetah** domain. To our knowledge, this is the first empirical study of the transferability

between domains with different dynamics and states of inversely learned rewards. Despite these differences, TraIRL shows strong performance in the target task with zero or one shot transfer, as shown in the last column of Table 2. Details of this experiment are provided in Appendix D.6. Furthermore, Appendix D.1 provides a semantic analysis of the abstract state space, showing that an abstraction trained on quadrupeds captures transferable structure useful for bipedal locomotion, rather than merely encoding peculiarities of the expert data such as joint angles.

### 5.1.2 BENEFIT OF ABSTRACTION AND ITS VISUALIZATION

To understand why TraIRL performs better, we aim to gain some insight into our novel abstraction concept. In Table 3, we report the 1-Wasserstein distance ($W_1$) between the abstracted state densities of the two source tasks and between the densities of the target and each source, $W_1(\rho^*_{\mathcal{T}^i}(z), \rho^*_{\mathcal{T}^t}(z))$, for the Half Cheetah and Ant. We compare these to the corresponding $W_1$ distances between the ground state densities. To obtain the $W_1$ between ground state densities, we train a variant of $f$-IRL that replaces the $f$-divergence with the $W_1$ (see Appendix A.2 of Ni et al. (2021)).

Table 3: Abstraction yields a smaller $W_1$ in the Half Cheetah and Ant, which is desirable.

|  | 1-Wasserstein Distance | |
|---|---|---|
|  | Abstractions | Ground |
| Source (rear disabled) and Source (front disabled) | **0.36** | 1.37 |
| Source (rear disabled) and Target (no disability) | **0.62** | 2.83 |
| Source (front disabled) and Target (no disability) | **0.55** | 2.10 |
| Source (Leg 1, 2 disabled) and Source (Leg 0, 3 disabled) | **0.33** | 1.78 |
| Source (Leg 1, 2 disabled) and Target (Leg 1, 3 disabled) | **0.71** | 2.82 |
| Source (Leg 1, 2 disabled) and Target (Leg 0, 2 disabled) | **0.79** | 2.97 |

For each comparison in Table 3, the abstracted state densities yield a consistently lower 1-Wasserstein distance compared to the ground state densities. This indicates that the abstraction reduces task-specific variability in the abstract state space, bringing the source and target closer in terms of their occupancy measures. In particular, the 1-Wasserstein distance between the abstract state densities of optimal policies in the source and target tasks, $W_1(\rho^*_{\mathcal{T}^i}(z), \rho^*_{\mathcal{T}^t}(z))$, corresponds to Eq. 8 in Theorem 2. A smaller distance implies a tighter bound on the transfer error $\epsilon$, and therefore reflects better generalization of the abstract state space across tasks.

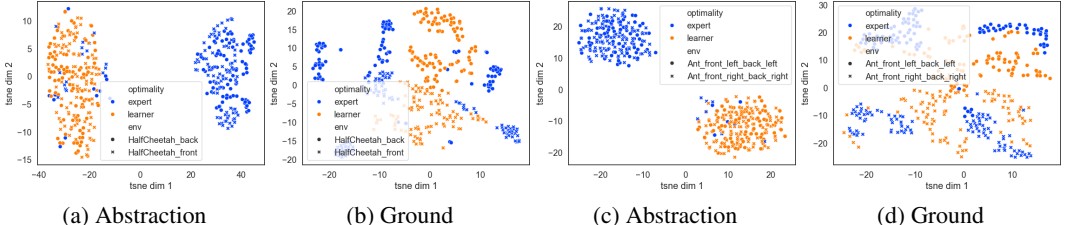

(a) Abstraction  (b) Ground  (c) Abstraction  (d) Ground

Figure 3: Visualization by t-SNE of (a) sampled abstractions and (b) ground states for Half Cheetah, and analogously for Ant (c,d) source environments. Details of these visualizations are in Appendix F

Next, Figure 3 visualizes the t-SNE (Van der Maaten & Hinton, 2008) of the abstract and ground states for our two source tasks in both Half Cheetah and Ant domains. There is a clear separation between the abstract states of the expert and the learner in Fig. 3a and Fig. 3c. More importantly, abstract states of the two source tasks are intermingled due to the abstraction, which is another indication of a dynamics-agnostic abstract state space. In contrast, Fig. 3b and Fig. 3d show that the ground states of the two source tasks appear in separate clusters. Furthermore, there is no single plane separating the embeddings of the expert and learner-induced states from their trajectories.

We also explored the sensitivity of TraIRL performance to hyperparameter values and conducted ablation studies on each part of TraIRL. The results are reported in Appendix D.3.

## 5.2 EVALUATION IN HUMAN-ROBOT ASSISTIVE GYM

To give an indication of the utility of TraIRL in the real-world, we illustrate its use in human-robot collaboration using the highly realistic Assistive Gym testbed (Erickson et al., 2019). Similar to MuJoCo, Assistive Gym is a physics-based simulation framework but for studying human-robot interaction and robotic assistance by the collaborative robot Sawyer. It consists of a suite of simulation environments for six tasks associated with activities of daily living such as itch scratching, bed bathing, drinking water, feeding, dressing, and arm manipulation.

For the source tasks, we select *FeedingSawyer*, a simulation environment in which the collaborative robot is tasked with feeding a disabled human. The two source tasks differ in the condition of the disabled human. The human is static in one, while the human has tremors in the other, which cause the target area to move resulting in a shifting goal. The robot's challenge is to adapt to these differing human conditions while executing the feeding task. Our target task is a different task, *ScratchItchSawyer*, in which Sawyer is tasked with scratching a disabled human's itch. Although this task differs from feeding in terms of its specific goal, it is also similar in that the robot must move its end effector precisely to a designated target area. This abstract information should be represented by the learned reward function. However, the distributions of goal states differ between source and target tasks. Thus, reward shaping is on top of the learned reward function $R_{\theta}$ in the target task by adding positive reward when goal states are reached: $\hat{R}(s) = R_{\theta}(\phi(s)) + \mathbb{I}(s \in \mathcal{G}) \cdot c$, where $\mathbb{I}(s \in \mathcal{G})$ is the indicator function, $\mathcal{G}$ is the set of goal states, and $c$ is a positive constant. An experiment motivating this reward shaping is reported in Appendix D.7. Note that this reward shaping is applied consistently to TraIRL as well as to all baseline methods to ensure a fair and comparable evaluation.

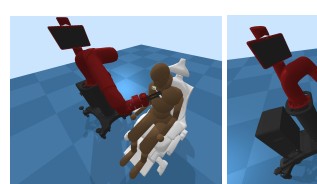

|  | Sources | | Target |
|---|---|---|---|
|  | Feeding task 1 | Feeding task 2 | Scratch itch |
| AIRL-ME | -6.35 ± 25.1 | 3.59 ± 17.3 | -17.32 ± 11.6 |
| RIME | 8.01 ± 6.0 | 8.66 ± 9.7 | -11.64 ± 4.2 |
| I2L | 9.32 ± 11.8 | 9.11 ± 11.4 | -10.07 ± 8.0 |
| **TraIRL** | 8.32 ± 7.9 | 9.56 ± 10.5 | **-3.82 ± 3.3** |
| Expert | 11.29 ± 5.3 | 12.77 ± 4.2 | -1.18 ± 5.7 |

(a) FeedingSawyer   (b) ScratchItchSawyer    (c)

Figure 4: Two tasks in the feeding domain (a) are used as sources to learn a reward function that is transferred to perform the task of scratching an itch (b). (c) Cumulative reward with standard deviation in the **Assistive Gym** environments. TraIRL has the highest reward in the target task.

We report the performance of TraIRL and the baselines in transferring the reward function inversely learned from the two feeding tasks, to learn how to scratch the person's itch. Table 4c gives the mean cumulative reward from forward RL in the target using the learned rewards. Notice that TraIRL yields the policy with the highest reward and close to the optimal, indicating transferable rewards. The transferred reward function exhibits a strong positive linear relationship with the ground-truth rewards, as indicated by a Pearson correlation coefficient of 0.86 ($p < .001$). This high correlation suggests that TraIRL's learning process effectively approximates the true reward function of *ScratchItchSawyer*, demonstrating the generalizability of the learned rewards in the target task. Among the baselines, I2L and RIME achieve comparative transfer, with performance close to each other on the source tasks but still weaker on the target. AIRL-ME lags further behind, showing limited ability to generalize. In contrast, TraIRL consistently outperforms all three, indicating that the abstracted reward function provides superior transferability.

## 6 CONCLUSION

TraIRL represents a significant advancement of IRL by introducing a principled approach to inversely learn transferable reward functions from demonstrations in multiple aligned tasks. The key contribution lies in its ability to extract invariant abstractions that model the structure intrinsic to multiple tasks, which makes them transferable to aligned target tasks. TraIRL's analytical properties delineate the transfer applicability of the abstracted rewards and the experiments validate the transferability in both formative and use-inspired contexts. Future work could investigate general ways of quickly fine-tuning the transferred reward function to improve its fit for a broader set of target tasks.

## 7 ETHICS STATEMENT

This work does not raise any specific ethical concerns. The research is purely methodological and focuses on algorithmic development without direct human or sensitive data involvement.

## 8 REPRODUCIBILITY STATEMENT

All hyperparameters, training details, and implementation choices for both TraIRL and the baseline methods are provided in Appendix C, ensuring the reproducibility of our results.

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

# Appendix

## Table of Contents

## A ALGORITHM

**Algorithm** 1 **demonstrates the training procedure for TraIRL**. During each iteration, trajectories are uniformly sampled from each source environment and policy, and added to the buffer (lines 6-7). The encoder $p_\phi$, decoders $q_{\psi^1}, ..., q_{\psi^n}$, discriminator $D_\omega$, and the reward function $\mathcal{R}_\theta$ are updated (line 8). Then, the learner policies $\pi_{\xi^1}, ..., \pi_{\xi^n}$ are updated in the source tasks $\mathcal{T}^1, ..., \mathcal{T}^n$ using the updated reward function $R_\theta$, respectively (line 9).

---

**Algorithm 1** TraIRL: *Training Phase*

---

**Require:** Expert trajectories $\tau_E^1, ..., \tau_E^n$; Source tasks: $\mathcal{T}^1, ..., \mathcal{T}^n$
1: Initialize learner policies $\pi_{\xi^1}, ..., \pi_{\xi^n}$; Trajectory buffer $B$; Discriminator $D_\omega$; Reward function $R_\theta$; encoder $p_\phi$; and decoders $q_{\psi^1}, ..., q_{\psi^n}$
2: Add expert trajectories $\tau_E^1, ..., \tau_E^n$ into trajectory buffer $B$
3: **while** $\pi_{\xi^1}, ..., \pi_{\xi^n}$ continue improving within $k$ steps **do**
4:     **for** task $i$ in $1, ..., n$ **do**
5:         Collect state-only trajectories $\tau^i = (s_0, ..., s_T)$
6:         Add trajectories $\tau^i$ into trajectory buffer $B$
7:     Uniformly sample trajectories $\tau$ from Buffer $B$
8:     Update $R_\theta$; $D_\omega$; $p_\phi$; and $q_{\psi^1}, ..., q_{\psi^n}$ using $\tau$ by Eq. 6
9:     Update $\pi_{\xi^1}, ..., \pi_{\xi^n}$ using $R_\theta$

---

---

**Algorithm 2** TraIRL: *Transfer Testing Phase*

---

**Require:** Reward function $R_\theta$ learned by Algorithm 1, Target task $\mathcal{T}^t$
1: Initialize policy $\pi_\xi$
2: **while** $\pi_\xi$ continues improving within $k$ steps **do**
3:     Update $\pi_\xi$ only using the learned reward function $R_\theta$ in target task $\mathcal{T}$

---

## B PROOF OF THEOREMS

### B.1 ANALYTICAL GRADIENT OF TraIRL (THEOREM 1)

*Proof.* In this section, we derive the analytic gradient of the proposed TraIRL (Theorem 1). From (Ni et al., 2021), we have the following equations:

$$\frac{d\rho_L(s)}{d\theta} = \int \frac{d\rho_L(s)}{dR_\theta(s^*)} \frac{dR_\theta(s^*)}{d\theta} ds^*$$

$$= \frac{1}{Z} \int p(\tau) e^{\sum_{t=1}^T R_\theta(s_t)} \eta_\tau(s) \sum_{t=1}^T \frac{dR_\theta(s_t)}{d\theta} d\tau - T\rho_L(s) \int \rho_L(s^*) \frac{dR_\theta(s^*)}{d\theta} ds^*, \tag{10}$$

where $Z$ is the normalization constant, and $\eta_\pi(s)$ denotes the number of times a state occurs in a trajectory $\tau$.

The joint distribution over states and abstracted states is defined as:

$$p(z, s) = p_\phi(z|s)\rho(s), \tag{11}$$

where $p_\phi(z|s)$ is the encoder parameterized by $\phi$.

The marginal distribution of the abstraction $z$, denoted as the abstract state density $\rho(z)$, is obtained by integrating out the state $s$ from Eq. 11:

$$\rho(z) = \int_{\mathcal{S}} p(z, s) ds$$

$$= \int_{\mathcal{S}} p_\phi(z|s)\rho(s) ds.$$

When operating in the abstracted state space, Eq. 10 becomes:

$$
\begin{aligned}
\frac{d\rho_L(z)}{d\theta} &= \int \frac{d\rho_L(z)}{dR_\theta(z^*)} \frac{dR_\theta(z^*)}{d\theta} dz^* \\
&= \frac{1}{Z} \int p(\tau) e^{\sum_{t=1}^{T} R_\theta(z_t)} \eta_\tau(z) \sum_{t=1}^{T} \frac{dR_\theta(z_t)}{d\theta} d\tau - T\rho_L(z) \int \rho_L(z^*) \frac{dR_\theta(z^*)}{d\theta} dz^*,
\end{aligned}
\tag{12}
$$

When optimizing the abstract state density matching objective between the expert density $\rho_E(z)$ and the learner density $\rho_L(z)$ in the $i$-th source environment, we measure their discrepancy using 1-Wasserstein distance. The objective is formulated as:

$$
\begin{aligned}
\mathcal{L}_{\mathcal{F}}(\boldsymbol{\theta}) &= \sum_{i=1}^{n} W_1(\rho_E(z), \rho_L(z)) \\
&= \sum_{i=1}^{n} \sup_{||f||_L \leq 1} \left| \mathbb{E}_{z \sim \rho_E(z)}[f(z)] - \mathbb{E}_{z \sim \rho_L(z)}[f(z)] \right| \\
&= \sum_{i=1}^{n} \max_{D_{\boldsymbol{\omega}}} \mathbb{E}_{z \sim \rho_E(z)}[D_{\boldsymbol{\omega}}(z)] - \mathbb{E}_{z \sim \rho_L(z)}[D_{\boldsymbol{\omega}}(z)] \\
&= \sum_{i=1}^{n} \max_{D_{\boldsymbol{\omega}}} \int_{\bar{\mathcal{S}}} D_{\boldsymbol{\omega}}(z) \rho_E(z) dz - \int_{\bar{\mathcal{S}}} D_{\boldsymbol{\omega}}(z) \rho_L(z) dz.
\end{aligned}
$$

The objective is derived using the Kantorovich–Rubinstein duality, which reformulates the 1-Wasserstein distance as a supremum over all 1-Lipschitz functions. To approximate this function, we introduce a discriminator $D_{\boldsymbol{\omega}}(z)$ and express the optimization as a maximization of the expected difference between expert and learner distributions. To ensure that $D_{\boldsymbol{\omega}}(z)$ satisfies the 1-Lipschitz constraint required by the duality, we apply a gradient penalty, which also stabilizes optimization while preserving theoretical correctness.

The gradient of the objective, Eq. 4, w.r.t $\boldsymbol{\theta}$ is derived as:

$$
\nabla_{\boldsymbol{\theta}} \mathcal{L}_{\mathcal{F}}(\boldsymbol{\theta}) = -\sum_{i=1}^{n} \int_{\bar{\mathcal{S}}} D_{\boldsymbol{\omega}}(z) \nabla_{\boldsymbol{\theta}} \rho_L(z) dz
\tag{13}
$$

Substituting the gradient of abstract state density $\rho_L(z)$ w.r.t $\theta$ with Eq.12, we have:

$$
\begin{aligned}
\nabla_{\boldsymbol{\theta}} \mathcal{L}_{\mathcal{F}}(\boldsymbol{\theta}) &\propto \frac{1}{T} \sum_{i=1}^{n} \int \rho_L(\tau^i) \sum_{t=1}^{T} D_{\boldsymbol{\omega}}(z_t) \sum_{t=1}^{T} \frac{dR_{\boldsymbol{\theta}}(z_t)}{d\boldsymbol{\theta}} d\tau^i \\
&\quad - T \sum_{i=1}^{n} \int_{\bar{\mathcal{S}}} D_{\boldsymbol{\omega}}(z) \rho_L(z) \left( \int_{\bar{\mathcal{S}}} \rho_L(z^*) \frac{dR_\theta(z^*)}{d\boldsymbol{\theta}} dz^* \right) dz \\
&= \frac{1}{T} \sum_{i=1}^{n} \mathbb{E}_{\tau \sim \rho_L(\tau^i)} \left[ \sum_{t=1}^{T} D_{\boldsymbol{\omega}}(z) \right] \sum_{t=1}^{T} \frac{dR_{\boldsymbol{\theta}}(z_t)}{d\boldsymbol{\theta}} \\
&\quad - T \sum_{i=1}^{n} \mathbb{E}_{z \sim \rho_L(z)}[D_{\boldsymbol{\omega}}(z)] \mathbb{E}_{z \sim \rho_L(z)} \left[ \frac{dR_{\boldsymbol{\theta}}(z)}{d\boldsymbol{\theta}} \right].
\end{aligned}
\tag{14}
$$

To gain further intuition about this equation, we can express all the expectations in terms of trajectories:

$$\nabla_{\boldsymbol{\theta}} \mathcal{L}_{\mathcal{F}}(\boldsymbol{\theta}) \propto \frac{1}{T} \sum_{i=1}^{n} \left( \sum_{t=1}^{T} D_{\boldsymbol{\omega}}(z_t) \sum_{t=1}^{T} \nabla_{\boldsymbol{\theta}} R_{\boldsymbol{\theta}}(z_t) \right.$$

$$\left. - \mathbb{E}_{\rho_L(\tau^i)} \left[ \sum_{t=1}^{T} D_{\boldsymbol{\omega}}(z_t) \right] \mathbb{E}_{\rho_L(\tau^i)} \left[ \sum_{t=1}^{T} \nabla_{\boldsymbol{\theta}} R_{\boldsymbol{\theta}}(z_t) \right] \right)$$

$$\propto \sum_{i=1}^{n} \sum_{t=1}^{T} \mathrm{cov}_{\tau \sim \rho_L(\tau^i)} \left( D_{\boldsymbol{\omega}}(z_t), \nabla_{\boldsymbol{\theta}} R_{\boldsymbol{\theta}}(z_t) \right).$$

$$= \sum_{i=1}^{n} \mathrm{cov}_{z \sim \rho_L(z)} \left( D_{\boldsymbol{\omega}}(z), \nabla_{\boldsymbol{\theta}} R_{\boldsymbol{\theta}}(z) \right).$$

$$= \sum_{i=1}^{n} \mathrm{cov}_{s \sim \rho_L(s^i), z \sim p_{\boldsymbol{\phi}}(z|s)} \left( D_{\boldsymbol{\omega}}(z), \nabla_{\boldsymbol{\theta}} R_{\boldsymbol{\theta}}(z) \right). \tag{15}$$

When operating in high-dimensional observation domains, a significant impediment arises if the state visitation distributions induced by the learner's current policy substantially diverge from the expert demonstration trajectories. Under such conditions of distributional mismatch, we empirically find that the gradient signal derived from our proposed objective function, Eq. 15, provides limited supervisory information to guide the policy optimization process. We follow the technique introduced by (Finn et al., 2016), mixing the data samples from expert trajectories with the learner trajectories. The revised objective function is given in the following.

$$\nabla_{\boldsymbol{\theta}} \mathcal{L}_{\mathcal{F}}(\boldsymbol{\theta}) = \sum_{i=1}^{n} \mathrm{cov}_{s \sim \hat{\rho}(s^i), z \sim p_{\boldsymbol{\phi}}(z|s)} \left( D_{\boldsymbol{\omega}}(z), \nabla_{\boldsymbol{\theta}} R_{\boldsymbol{\theta}}(z) \right), \tag{16}$$

where $\hat{\rho}(s^i) = \frac{1}{2}(\rho_L(s^i) + \rho_E(s^i))$. $\qquad\square$

### B.2 PROOF OF THEOREM 2

*Proof.* We begin by noting that the 1-Wasserstein distance is a **metric** and therefore satisfies the triangle inequality:

$$W_1(\rho_1, \rho_3) \le W_1(\rho_1, \rho_2) + W_1(\rho_2, \rho_3).$$

Apply the triangle inequality to the three distributions:

- $\rho_{\mathcal{T}^t}^*(z)$ - abstract state density of the expert in the target task.

- $\rho_{\mathcal{T}^i}^*(z)$ - abstract state density of the expert in the $i$-th source task.

- $\rho_{\mathcal{T}^i}(z)$ - abstract state density of the learner in the $i$-th source task.

By the triangle inequality:

$$W_1(\rho_{\mathcal{T}^t}^*(z), \rho_{\mathcal{T}^i}(z)) \le W_1(\rho_{\mathcal{T}^t}^*(z), \rho_{\mathcal{T}^i}^*(z)) + W_1(\rho_{\mathcal{T}^i}^*(z), \rho_{\mathcal{T}^i}(z))$$
$$\le \alpha\epsilon + (1-\alpha)\epsilon$$
$$= \epsilon.$$

This satisfies the condition in Def 2, which requires that the abstract state density induced by the learned reward function $R_{\boldsymbol{\theta}}$ in a source task is close (within $\epsilon$) to the optimal policy's abstract state density in the target task.

Hence, $R_{\boldsymbol{\theta}}$ is transferable. $\qquad\square$

### B.3 DYNAMICS DISENTANGLED STATE-ONLY REWARD FUNCTION

In this section, we follow the derivations and definitions of Fu et al. (2018) to establish that TraIRL learns disentangled reward functions. For completeness, we restate the key definitions and theorems here. We first define the induced ground-level reward function using the abstract reward function in TraIRL.

**Definition 3** (Induced ground-level reward function). *Let $\phi : \mathcal{S} \to \mathcal{Z}$ be TraIRL's abstraction function and let $r_{abs} : \mathcal{Z} \to \mathbb{R}$ be TraIRL's abstract reward function. Define the induced ground-level reward function*

$$r_\phi(s) := r_{abs}(\phi(s)).$$

Then we borrow the definition of "disentangled rewards" from Fu et al. (2018).

**Definition 4** (Disentangled rewards). *A reward function $r'(s, a, s')$ is (perfectly) disentangled with respect to a ground-truth reward $r(s, a, s')$ and a set of dynamics $\mathcal{T}$ such that under all dynamics $T \in \mathcal{T}$, the optimal policy is the same: $\pi^*_{r',T}(a \mid s) = \pi^*_{r,T}(a \mid s)$.*

Disentangled rewards can be informally understood as reward functions that induce the same optimal policy as the ground truth reward under any admissible dynamics. To demonstrate how TraIRL recovers such a reward, we first recall the definition of the decomposability condition.

**Definition 5** (Decomposability condition). *Two states $s_1, s_2$ are defined as "1-step linked" under a dynamics or transition distribution $T(s' \mid a, s)$ if there exists a state $s$ that can reach $s_1$ and $s_2$ with positive probability in one time step. Also, we define that this relationship can transfer through transitivity: if $s_1$ and $s_2$ are linked, and $s_2$ and $s_3$ are linked, then we also consider $s_1$ and $s_3$ to be linked.*
*A transition distribution $T$ satisfies the decomposability condition if all states in the MDP are linked with all other states.*

Theorem 3 and 4 formalize that TraIRL recovers reward functions disentangled from the dynamics.

**Theorem 3.** *Let $r(s)$ be a ground-truth reward, and $T$ be a dynamics model satisfying the decomposability condition. Suppose IRL recovers a state-only reward $r'(s)$ such that it produces an optimal policy in $T$:*

$$Q^*_{r',T}(s, a) = Q^*_{r,T}(s, a) - f(s).$$

*Then, $r'(s)$ is disentangled with respect to all dynamics.*

*Proof.* Refer to Theorem 5.1 in Fu et al. (2018). $\square$

**Theorem 4.** *If a reward function $r'(s, a, s')$ is disentangled for all dynamics functions, then it must be state-only, i.e. if for all dynamics $T$,*

$$Q^*_{r,T}(s, a) = Q^*_{r',T}(s, a) + f(s) \quad \forall s, a.$$

*Then $r'$ is only a function of state.*

*Proof.* Refer to Theorem 5.2 in Fu et al. (2018). $\square$

Next, we demonstrate an example where the decomposability condition is not satisfied, whereas in the TraIRL, a disentangled reward function can still be learned. Consider the following 3-state MDP with deterministic dynamics and starting state A:

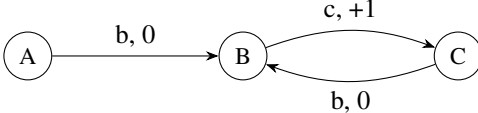

State A cannot be reached from any other states in the MDP, thus, the decomposability condition is not satisfied. However, if there exists an abstraction $\phi$, where $\phi(A) = \phi(B) = Z$, then the abstract MDP becomes:

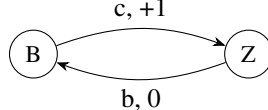

Thus, the new abstract MDP satisfies the decomposability condition and there exists a disentangled reward function as list above.

## C  TRAINING DETAILS AND HYPERPARAMETERS

In this section, we show the comprehensive training details and hyperparameters. We use Soft Actor-Critic (SAC) as our Maximum Entropy Reinforcement Learning (MaxEnt RL) algorithm due to its efficient exploration, stability in continuous control tasks, and improved sample efficiency. By maximizing both cumulative reward and entropy, SAC promotes diverse and robust policies. For implementation, we use the SAC provided by the widely adopted Python library Stable-Baselines 3.

To generate expert demonstrations, we first train SAC agents with 5 different random seeds in each source domain until convergence. The hyperparameters used for SAC training are listed in Table 4. The unlisted hyperparameter remains the default setting in Stable-Baselines 3. After convergence, we collect 50 expert trajectories from each source domain. These expert trajectories are then used for training the transferable reward function via TraIRL as well as other baseline methods.

Table 4: Hyperparameter setting of SAC.

|  | Ant | HalfCheetah | FeedingSawyer | ScratchItchSawyer |
|---|---|---|---|---|
| Learning rate | $3e^{-4}$ | $3e^{-4}$ | $3e^{-4}$ | $3e^{-4}$ |
| Gamma | 0.99 | 0.99 | 0.99 | 0.99 |
| Batch size | 256 | 256 | 256 | 256 |
| Net arch | $[400, 300]$ | $[400, 300]$ | $[400, 400]$ | $[400, 400]$ |
| Buffer size | $1,000,000$ | $1,000,000$ | $1,000,000$ | $100,000$ |
| Action noise | $\mathcal{N}(0, 0.2)$ | $\mathcal{N}(0, 0.2)$ | $\mathcal{N}(0, 0.2)$ | $\mathcal{N}(0, 0.25)$ |

Next, we begin training TraIRL and baselines. The hyperparameters used in each source domain are listed in Table 5-8. Notably, the SAC in TraIRL adopts the same hyperparameters specified in Table 4. The net arch represents the dimensions of the model for each layer, excluding the output layer. The reward function produces a single scalar value, which is activated by a Tanh function. The update step refers to the number of gradient updates performed during each iteration of the training process. All baselines are evaluated using their default hyperparameters. The only modification is for RIME, where the input to the discriminator is restricted to state-only features to ensure a fair comparison with our approach.

After training TraIRL, we obtain a trained transferable reward function, which is then applied to the target task by replacing the original reward function. Consequently, when the agent interacts with the target task, it only has access to the trained reward function. We continue to use SAC with the hyperparameters specified in Table 4 for policy optimization in the target domain.

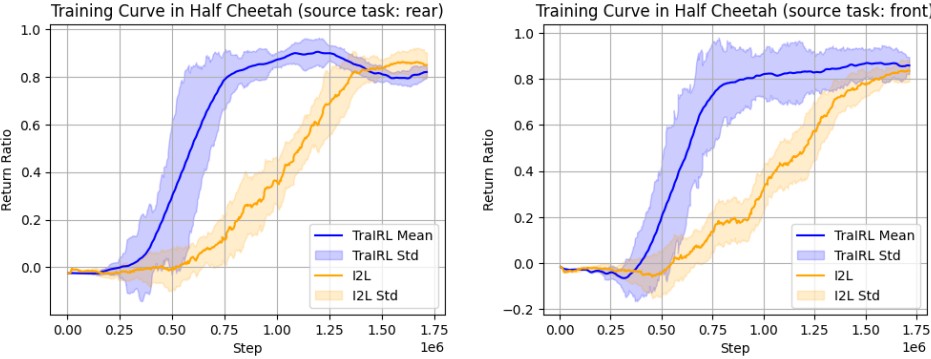

(a) Training curve in Half Cheetah (rear disabled). (b) Training curve in Half Cheetah (front disabled)

Figure 5: Smoothed training curve for Half Cheetah in two source tasks. AIRL-ME and $f$-IRL perform poorly in the experiments and are therefore excluded from the comparison.

Table 5: Hyperparameter setting of TraIRL (Algorithm 1).

|  | Ant | HalfCheetah | FeedingSawyer |
|---|---|---|---|
| $\lambda_{\mathrm{GP}}$ | 10 | 10 | 0.1 |
| $\lambda_{\mathcal{D}}$ | 0.1 | 0.1 | 0.1 |
| $\lambda_{\mathrm{VAE}}$ | 1.0 | 1.0 | 1.0 |
| $\lambda_{\mathrm{WGAN}}$ | 1.0 | 1.0 | 1.0 |
| $\lambda_{\mathcal{F}}$ | 1.0 | 1.0 | 1.0 |
| Reward Function Hyperparameter | | | |
| Learning Rate | 3e-4 | 3e-4 | 5e-4 |
| Batch Size | 256 | 256 | 256 |
| Weight Decay | 1e-3 | 1e-3 | 1e-3 |
| Net arch | [16, 16] | [16, 16] | [16, 16] |
| Activation | Tanh | Tanh | Tanh |
| Reward Update Steps | 10 | 10 | 10 |
| VAE and Discriminator Hyperparameter | | | |
| Learning Rate | 3e-4 | 3e-4 | 5e-4 |
| Batch Size | 256 | 256 | 256 |
| Weight Decay | 1e-3 | 1e-3 | 1e-3 |
| Encoder Net Arch | [32, 32, 32] | [32, 32, 32] | [16, 16] |
| Encoder Activation | Tanh | Tanh | Tanh |
| Abstraction Dimension | 16 | 10 | 4 |
| Decoder Net Arch | [64, 64, 64] | [64, 64, 64] | [16, 16, 16] |
| Decoder Activation | Tanh | Tanh | Tanh |
| VAE Update Steps | 10 | 10 | 10 |
| Discriminator Net Arch | [32, 32] | [32, 32] | [16, 16] |
| Discriminator Activation | Tanh | Tanh | Tanh |
| Disc Update Steps | 10 | 10 | 10 |

Table 6: Hyperparameter setting of AIRL-ME.

|  | Ant | HalfCheetah | FeedingSawyer |
|---|---|---|---|
| $g_\theta(s)$ network | [64, 64, 64] | [64, 64, 64] | [64, 64, 64] |
| $h_{\phi^i}(s)$ network | [64, 64, 64] | [64, 64, 64] | [64, 64, 64] |
| Learning rate | 3e-4 | 3e-4 | 3e-4 |
| Batch size | 256 | 256 | 256 |
| Weight Decay | 1e-3 | 1e-3 | 1e-3 |
| Activation | Tanh | Tanh | Tanh |
| Discriminator gradient steps | 10 | 10 | 10 |

Table 7: Hyperparameter setting of RIME.

|  | Ant | HalfCheetah | FeedingSawyer |
|---|---|---|---|
| Policy network | [400, 300] | [400, 300] | [400, 300] |
| Policy algorithm, lr, gradient-steps | PPO, 3e-4, 5 | PPO, 3e-4, 5 | PPO, 3e-4, 5 |
| Discriminator network | [100, 100] | [100, 100] | [100, 100] |
| Input to the discriminator | State-only | State-only | State-only |
| Discriminator gradient-steps | 5 | 5 | 5 |
| Gradient penalty term | 10 | 10 | 10 |
| Batch size | 256 | 256 | 256 |
| Activation | Tanh | Tanh | Tanh |

# D    EXTRA EXPERIMENTS AND CLARIFICATIONS

## D.1    SEMANTIC ANALYSIS OF ABSTRACTED STATES

In this section, we conduct a semantic analysis of the abstracted states, which analyzes the effect of each ground state dimension on the learned abstract state. Algorithm 3 demonstrates the experiment procedure for semantic analysis.

Table 8: Hyperparameter setting of I2L.

|  | Ant | HalfCheetah | FeedingSawyer |
|---|---|---|---|
| Wasserstein critic network | [64, 64, 64] | [64, 64, 64] | [64, 64, 64] |
| Discriminator network | [64, 64, 64] | [64, 64, 64] | [64, 64, 64] |
| Policy network | [400, 300] | [400, 300] | [400, 300] |
| Wasserstein critic optimizer, lr, gradient-steps | Adam, 5e-5, 20 | Adam, 5e-5, 20 | Adam, 5e-5, 20 |
| Discriminator optimizer, lr, gradient-steps | Adam, 3e-4, 5 | Adam, 3e-4, 5 | Adam, 3e-4, 5 |
| Policy algorithm, lr | PPO, 1e-4 | PPO, 1e-4 | PPO, 1e-4 |
| Batch size | 256 | 256 | 256 |
| Activation | Tanh | Tanh | Tanh |

Table 9: Source codes of baselines.

| Algorithm | URL |
|---|---|
| AIRL-ME | https://github.com/Ojig/Environment-Design-for-IRL |
| RIME | https://github.com/JongseongChae/RIME |
| I2L | https://github.com/tgangwani/RL-Indirect-imitation |

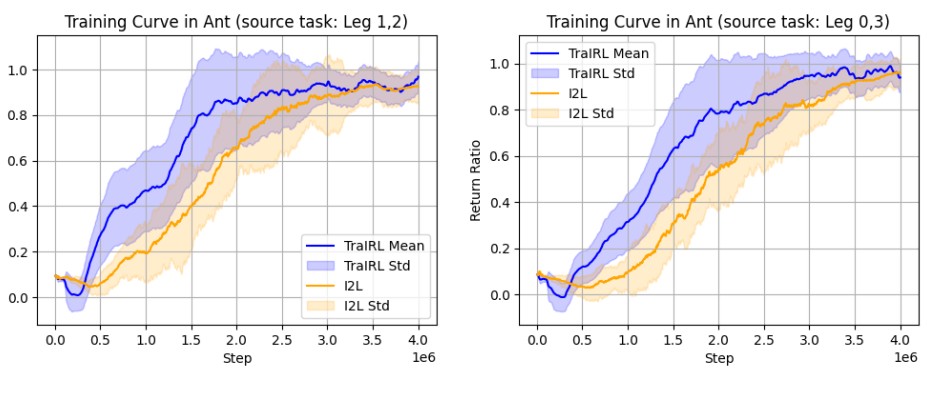

(a) Training curve in Ant (Leg 1 & 2).  (b) Training curve in Ant (Leg 0 & 3).

Figure 6: Smoothed training curve for Ant in two source tasks. AIRL-ME and $f$-IRL perform poorly in the experiments and are therefore excluded from the comparison.

---

**Algorithm 3** Semantic Analysis
---
**Require:** Learned Encoder $p_\phi$, Trajectory buffer $B$.
1: Sample states $s$ from Trajectory buffer $B$.
2: **for** each dimension $d$ of the states **do**
3:    Create two perturbed states $s^+, s^-$ by adding and subtracting 1.0 to dimension $d$.
4:    Encode $s^+, s^-$ using the learned encoder $p_\phi$.
5:    Compute Cosine similarity between the resulting abstracted states: $cos\_similarity(p_\phi(s^+), p_\phi(s^-))$.

---

Figure 7 shows the absolute value of cosine similarity. A high cosine similarity indicates the abstract representation is insensitive to perturbation in that dimension, implying it may be semantically less important. Conversely, a low similarity suggests that the dimension is critical in shaping the latent abstraction. For the Ant task, the lowest similarity consistently appears at dimension 13 across all tasks, indicating its high semantic importance in the abstract representation. According to Gymnasium documentation[1], dimension 13 corresponds to the x-velocity of the torso, i.e., the forward movement. Similarly, for the Half Cheetah task[2], the lowest similarity appears at dimension 8, corresponding to the velocity of the x-coordinate of the front tip. In both cases, the critical feature corresponds to forward speed, which emerges as a common and invariant component across tasks. These findings

---

[1] https://gymnasium.farama.org/environments/mujoco/ant/#observation-space
[2] https://gymnasium.farama.org/environments/mujoco/half_cheetah/#observation-space

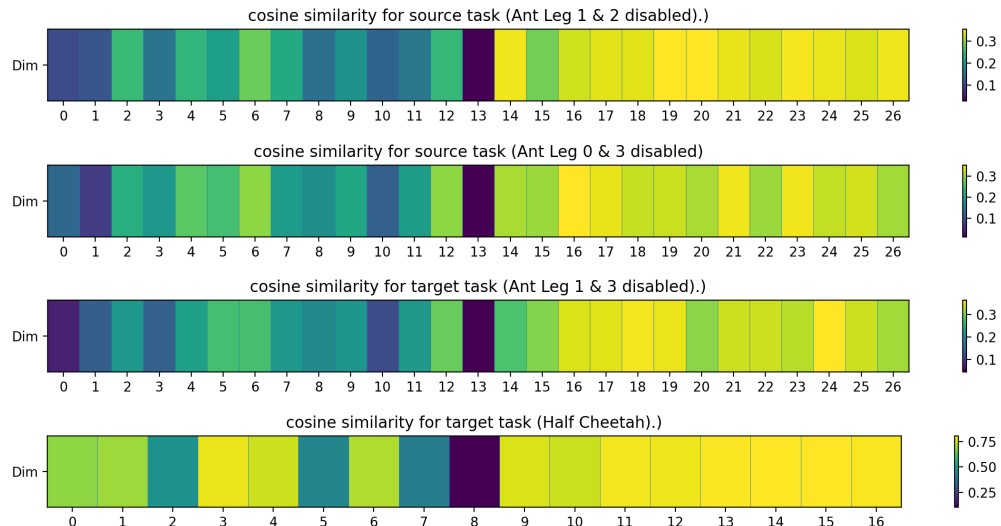

Figure 7: Visualization result for the semantic analysis via Cosine similarity.

support the claim that the learned abstraction not only captures semantically relevant information but also emphasizes task-invariant features that are essential for generalization.

## D.2 INTEGRATE ABSTRACT STATES INTO OTHER IRL ALGORITHMS

In this section, we evaluate whether the abstraction representation $\phi$ can be plugged into other IRL algorithms.

Table 10: Comparison of TraIRL, AIRL with abstraction, and SFM with abstraction as the feature function on source and target tasks.

| | Sources | | Targets | |
|---|---|---|---|---|
| | Source 1 | Source 2 | Target 1 | Target 2 |
| AIRL with abstraction | $2579.61 \pm 68.3$ | $\mathbf{2966.92 \pm 67.8}$ | $2271.65 \pm 127.7$ | $2766.01 \pm 132.7$ |
| SFM with abstraction | $\mathbf{3002.88 \pm 51.2}$ | $2851.09 \pm 71.4$ | $72.09 \pm 18.3$ | $52.78 \pm 10.1$ |
| TraIRL | $2714.18 \pm 35.9$ | $2936.52 \pm 95.5$ | $\mathbf{2917.92 \pm 84.5}$ | $\mathbf{3156.54 \pm 63.1}$ |
| Expert | $3312.12 \pm 304.3$ | $3303.99 \pm 341.0$ | $3369.05 \pm 216.8$ | $3590.57 \pm 158.2$ |

As shown in Table 10, both AIRL with abstraction and SFM with abstraction achieve higher returns than TraIRL on the source tasks, demonstrating their ability to fit the training distributions more closely. However, these methods fail to preserve this advantage in the target tasks: SFM collapses almost entirely, and AIRL suffers a large drop in performance. In contrast, TraIRL maintains strong generalization across targets. Compared to AIRL, $f$-IRL benefits from its separate reward function network, independent of the discriminator. The reward function is trained to maximize covariance with the discriminator's output, effectively serving as a distillation process. This distillation enables the reward function to capture transferable information rather than overfitting to the source distributions, thereby improving generalization. For SFM with abstraction, the successor feature function is updated using the temporal consistency loss

$$\mathbb{E}_{(s,a,s') \sim \mathcal{D},\, a' \sim \pi_\mu(\cdot|s')} \left[ \parallel \phi(s) + \psi_\theta(s', a') - \psi(s, a) \parallel_2^2 \right],$$

where $\mathcal{D}$ is the buffer, $\phi(s)$ is the base feature function, and $\psi_\theta(s, a)$ is the successor feature function. Because this update depends directly on the current policy and state distribution, the learned successor features are tied to the specific dynamics and current policy. When the policy or state distribution shifts, as in the target tasks, the successor feature function becomes inaccurate. Since we do not retrain the successor features in the transfer setting, this mismatch results in poor generalization performance on unseen tasks.

## D.3 HYPERPARAMETER SENSITIVITY ANALYSIS

In this section, we evaluate the sensitivity of TraIRL's performance on the coefficients $\lambda_{\text{GP}}$ and $\lambda_{\mathcal{D}}$. The coefficient $\lambda_{\text{GP}}$ controls the magnitude of the gradient penalty when updating the discriminator (Eq. 3), while $\lambda_{\mathcal{D}}$ regulates the strength of the regularization term when updating the encoder (Eq. 2).

**Hyperparameter $\lambda_{\textbf{GP}}$** The coefficient $\lambda_{\text{GP}}$ controls the magnitude of the gradient penalty when

Table 11: Mean cumulative reward with standard deviation in the **Half Cheetah** domain. $\lambda_{\text{GP}} = 10$ yields the highest cumulative reward in the target domain.

|  | Sources | | Target |
|---|---|---|---|
|  | Run (rear disabled) | Run (front disabled) | Run (no disability) |
| $\lambda_{\text{GP}} = 1$ | $4{,}646.12 \pm 40.6$ | $4{,}602.13 \pm 24.1$ | $3{,}228.27 \pm 185.5$ |
| $\lambda_{\text{GP}} = 5$ | $4{,}498.74 \pm 59.5$ | $4{,}313.89 \pm 54.4$ | $3{,}989.06 \pm 61.6$ |
| $\lambda_{\text{GP}} = 10$ | $4{,}404.07 \pm 57.6$ | $4{,}359.35 \pm 99.2$ | $\textbf{5{,}835.11} \pm \textbf{74.0}$ |
| $\lambda_{\text{GP}} = 100$ | $172.69 \pm 191.9$ | $155.66 \pm 159.5$ | $150.41 \pm 102.6$ |
| Expert | $5{,}052.3 \pm 25.4$ | $5{,}499.07 \pm 156.1$ | $6{,}420.38 \pm 37.9$ |

updating the discriminator (Eq. 3). Table 11 presents the mean cumulative reward with standard deviation for different values of $\lambda_{\text{GP}}$ in the Half Cheetah domain. The results demonstrate that $\lambda_{\text{GP}}$ significantly influences performance across both the source and target environments. Notably, when $\lambda_{\text{GP}} = 10$, the model achieves the highest cumulative reward in the target domain, indicating that this setting balances the regularization effect of the gradient penalty. In contrast, setting $\lambda_{\text{GP}}$ too low ($\lambda_{\text{GP}} = 1$) results in suboptimal performance, likely due to insufficient constraint on the discriminator. Recall that the 1-Lipschitz constraint is a crucial condition for the discriminator to approximate the 1-Wasserstein distance. If $\lambda_{\text{GP}}$ is too small, it may lead to a violation of this constraint, preventing an accurate approximation of the 1-Wasserstein distance. Consequently, Theorem 2 is not satisfied, undermining the theoretical guarantees of TraIRL. Conversely, an excessively large $\lambda_{\text{GP}}$ ($\lambda_{\text{GP}} = 100$) drastically degrades performance across all environments, suggesting that an overly strong gradient penalty hinders learning by excessively constraining the discriminator. These findings emphasize the importance of tuning $\lambda_{\text{GP}}$ to maintain theoretical validity while achieving optimal generalization in the target domain.

**Hyperparameter $\lambda_{\mathcal{D}}$**

Table 12: Mean cumulative reward with standard deviation in the **Half Cheetah** domain. $\lambda_{\mathcal{D}} = 0.1$ yields the highest cumulative reward in the target domain.

|  | Sources | | Target |
|---|---|---|---|
|  | Run (rear disabled) | Run (front disabled) | Run (no disability) |
| $\lambda_{\mathcal{D}} = 0.05$ | $4{,}323.23 \pm 29.58$ | $4{,}471.05 \pm 56.21$ | $4{,}541.09 \pm 96.30$ |
| $\lambda_{\mathcal{D}} = 0.1$ | $4{,}404.07 \pm 57.6$ | $4{,}359.35 \pm 99.2$ | $\textbf{5{,}835.11} \pm \textbf{74.0}$ |
| $\lambda_{\mathcal{D}} = 0.25$ | $4{,}430.26 \pm 62.7$ | $4{,}234.33 \pm 84.0$ | $4{,}548.23 \pm 44.5$ |
| $\lambda_{\mathcal{D}} = 0.5$ | $3{,}806.92 \pm 85.3$ | $3{,}888.79 \pm 52.5$ | $4{,}088.95 \pm 83.8$ |
| Expert | $5{,}052.25 \pm 25.4$ | $5{,}499.07 \pm 156.1$ | $6{,}420.38 \pm 37.9$ |

$\lambda_{\mathcal{D}}$ regulates the strength of the regularization term when updating the encoder (Eq. 2), leading to a compact and generalizable abstracted state space. Table 12 reports the mean cumulative reward across different values of $\lambda_{\mathcal{D}}$ in the Half Cheetah domain. At $\lambda_{\mathcal{D}} = 0.1$, the model achieves the highest cumulative reward in the target environment ($4745.59 \pm 48.56$), indicating that this value provides a balance for learning effective latent representations. Lowering $\lambda_{\mathcal{D}}$ to 0.05 slightly reduces performance in the target domain, suggesting that insufficient regularization may lead to suboptimal feature extraction. Increasing $\lambda_{\mathcal{D}}$ beyond 0.1 results in a noticeable performance degradation. At $\lambda_{\mathcal{D}} = 0.25$, the reward declines to $4548.23 \pm 44.49$, and at $\lambda_{\mathcal{D}} = 0.5$, it further drops to $4088.95 \pm 83.82$. This decline suggests that excessive regularization constrains the encoder, limiting its ability to adapt to the target task. The performance reduction is also observed in source tasks, indicating that overly strong regularization affects overall learning stability.

### D.4    INSUFFICIENCY OF SINGLE SOURCE TASK

In this section, we explore the insufficiency of a single-source task in training a transferable reward function. Recall that the objective of IRL is to match the state density or occupancy measure between the learner and expert policy. Therefore, if we only have a single source task, the learned reward function does not have to be generalized to any tasks except for the trained source task, even if the ground true reward function shares the same structure across tasks. As shown in Table 13, each reward function performs well on its corresponding training source task but generalizes poorly to others. This result highlights the necessity of using multiple diverse source tasks to capture a task-invariant reward structure for effective transfer learning.

Table 13: Single source task transfer learning experiments on Half Cheetah.

|  |  | **Target** | | |
|  |  | Rear Disabled | Front Disabled | Normal |
|---|---|---|---|---|
| **Source** | Rear Disabled | **4,867.19 ± 47.3** | 532.67 ± 312.2 | 3,158.79 ± 211.1 |
|  | Front Disabled | 831.98 ± 447.5 | **5,318.55 ± 83.9** | 3,351.9 ± 198.4 |
|  | Normal | 2,683.83 ± 412.7 | 3,017.66 ± 316.7 | **6,211.72 ± 42.0** |

### D.5    ADDITIONAL DETAILS ABOUT THE 1-WASSERSTEIN DISTANCE EXPERIMENTS

In this section, we detail the 1-Wasserstein distance experiments presented in Sec. 5.1.2. The distance in the abstracted state space is computed using TraIRL, while the corresponding distance in the ground state space is obtained from a variant of $f$-IRL in which the $f$-divergence is replaced by the 1-Wasserstein distance (see Appendix A.2 of Ni et al. (2021)). In both cases, the 1-Wasserstein distance is estimated using the discriminator from a WGAN trained on expert trajectories from two source tasks. Although expert policies for target tasks are typically unavailable in standard IRL or transfer learning settings, we assume access to them in this experiment to enable the computation of the 1-Wasserstein distance defined in Theorem 2, $W_1(\rho^*_{\mathcal{T}^t}(z), \rho^*_{\mathcal{T}^i}(z))$, between the abstracted occupancy measures of the target and source tasks. The distances reported in Table 14 are computed from the state densities of the **expert policy**, either in the abstracted state space or the ground state space.

Table 14: Abstraction yields a smaller $W_1$ in the Half Cheetah and Ant, which is desirable.

|  |  | **1-Wasserstein Distance** | |
|  |  | Abstractions | Ground |
|---|---|---|---|
| **HalfCheetah** | Source (rear disabled) and Source (front disabled) | **0.36** | 1.37 |
|  | Source (rear disabled) and Target (no disability) | **0.62** | 2.83 |
|  | Source (front disabled) and Target (no disability) | **0.55** | 2.10 |
| **Ant** | Source (Leg 1, 2 disabled) and Source (Leg 0, 3 disabled) | **0.33** | 1.78 |
|  | Source (Leg 1, 2 disabled) and Target (Leg 1, 3 disabled) | **0.71** | 2.82 |
|  | Source (Leg 1, 2 disabled) and Target (Leg 0, 2 disabled) | **0.79** | 2.97 |
|  | Source (Leg 0, 3 disabled) and Target (Leg 1, 3 disabled) | **0.81** | 3.00 |
|  | Source (Leg 0, 3 disabled) and Target (Leg 0, 2 disabled) | **0.75** | 2.89 |

### D.6    TRANSFER LEARNING FROM ANT TO HALF CHEETAH

In this section, we describe the details for performing transfer learning from the source domain (Ant) to the target domain (Half Cheetah). The primary challenge in this cross-domain transfer arises from the mismatch between the ground state spaces of the two environments. Specifically, the Ant domain has a 27-dimensional state space, whereas the Half Cheetah domain has a 17-dimensional state space. Although both domains primarily consist of joint angles and velocities, the ranges of these values differ significantly between the two robots. As a result, directly applying the encoder trained in the source domain to the target domain leads to a substantial distribution shift, undermining the effectiveness of learned representations and transferred rewards. To mitigate the distribution shift, we introduce two methods, **one-shot** transfer learning and **zero-shot** transfer learning. Fig. 8 shows the overview of transfer learning.

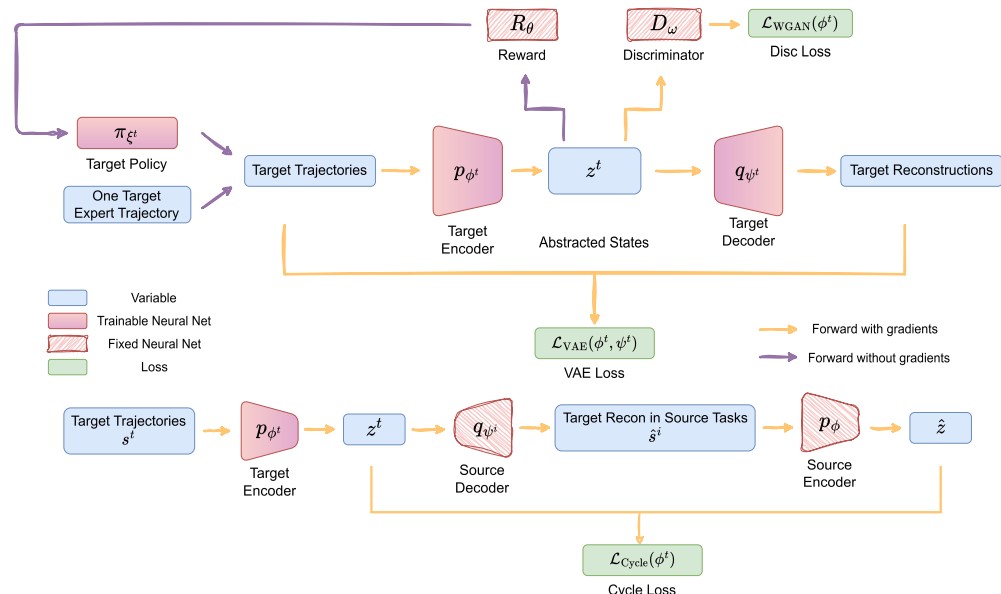

Figure 8: Overview of one-shot transfer learning in the target task.

**One-Shot Transfer Learning**   We first introduce one-shot transfer learning. Compared to training in the source domain, we have trained the reward function $R_{\boldsymbol{\theta}}$, the discriminator $D_{\omega}$, the encoder $p_{\phi}$ and the decoder $q_{\psi^i}$ for the source domain. Importantly, there is only **one** expert trajectory in the target domain, the so-called one-shot transfer learning. Specifically, we need to train an encoder $p_{\phi^t}$, a decoder $q_{\psi^t}$, and a policy $\pi_{\xi^t}$ for the target domain. The VAE loss function remains the same as the training in the source domain, except for the single decoder in the target domain.

$$\mathcal{L}_{\text{VAE}}(\phi^t, \psi^t) = \mathbb{E}_{z \sim p_{\boldsymbol{\phi}^t}(z^t | s^t)} \left[ \log q_{\psi^t}(s^t | z) \right] - \lambda_{\mathcal{D}} \, \mathcal{D}_{\text{KL}} \left( p_{\boldsymbol{\phi}^t}(z^t | s^t) \, \| \, p(z^t) \right). \quad (17)$$

Since the reward function $R_{\theta}$ has been trained, we no longer need the reward loss. In terms of discriminator loss, the discriminator has also been trained. Therefore, the discriminator loss now only updates the encoder parameters:

$$\begin{aligned}
\mathcal{L}_{\text{WGAN}}(\phi^t) = \; & \mathbb{E}_{z \sim p_{\boldsymbol{\phi}^t}(z|s), s \sim \rho_L(s^t)}[D_{\boldsymbol{\omega}}(z)] - \mathbb{E}_{z \sim p_{\boldsymbol{\phi}^t}(z|s), s \sim \rho_E(s^t)}[D_{\boldsymbol{\omega}}(z)] \\
& + \lambda_{\text{GP}} \, \mathbb{E}_{z \sim \hat{\rho}(z^t)} \left[ \left( \| \nabla_z D_{\boldsymbol{\omega}}(z) \|_2 - 1 \right)^2 \right] \Big),
\end{aligned} \quad (18)$$

Next, we introduce the novel cycle loss, illustrated in the overview in Fig. 8. The dimension of the abstracted state in the target domain is the same as that in the source domain. While the VAE loss and discriminator loss shape the abstract state space to be compact and optimality-aware, the cycle loss is designed to establish semantic alignment between the source and target domains. Specifically, it encourages consistency between the abstract state directly encoded from a target ground state and the abstract state obtained by first reconstructing that target abstracted state into the source domain and then encoding it. In doing so, the cycle loss ensures that abstract states from the target domain remain meaningfully aligned with those from the source, facilitating effective reward transfer despite differences in the ground state spaces.

$$\mathcal{L}_{\text{Cycle}}(\phi^t) = \mathcal{D}_{\text{KL}}\left( p_{\phi^t}(z^t | s^t) \, || \, p_{\phi}(\hat{z} | \hat{s}^i) \right), \quad (19)$$

where $\hat{s}^i \sim q_{\psi^i}(s^i | z^t)$.

The overall loss function is the linear combination of the three loss functions:

$$\mathcal{L}(\phi^t, \psi^t) = \lambda_{\text{VAE}} \mathcal{L}_{\text{VAE}} + \lambda_{\text{WGAN}} \mathcal{L}_{\text{WGAN}} + \lambda_{\text{Cycle}} \mathcal{L}_{\text{Cycle}} \quad (20)$$

**Zero-Shot Transfer Learning**   Next, we introduce the zero-shot transfer learning setting, which requires the specification of goal criteria for expert behavior. To guide policy learning in the absence of expert demonstrations in the target domain, we apply a variant of the Hindsight Experience Replay (HER) technique to the learner trajectories. Unlike the original HER, which relabels achieved states as goals, our variant replaces the failed goal conditions in the learner trajectories with the expert-specified goal criteria. This relabeling produces more informative optimality signals, allowing the agent to learn effectively from suboptimal trajectories that would otherwise provide little to no feedback. For instance, in our zero-shot setting for the target domain (Half Cheetah), the goal criterion is achieving a high forward velocity. In the learner trajectories, where the agent typically exhibits low forward speed, we relabel the corresponding state features by replacing the observed speed with a high value that reflects expert-level performance. This substitution allows the learner to receive feedback aligned with the desired goal, thereby facilitating learning despite the absence of expert demonstrations in the target domain.

The loss function of zero-shot transfer learning remains the same as Eq. 20.

Table 15: Transfer learning experiments on Half Cheetah.

|  | Half Cheetah (One-Shot) | Half Cheetah (Zero-Shot) |
| --- | --- | --- |
| Ant (Leg 1, 2) & (Leg 0, 3) (Source Tasks) | $5,378.78 \pm 61.7$ | $4,821.79 \pm 152.5$ |

### D.7   FINE-TUNING WITH REWARD SHAPING

In Sec. 5.2, we mention that reward shaping is applied to the learned reward function in the target task. Since the distributions of goal states differ between the source and target tasks, the learned reward alone is insufficient to guide the agent toward the new goal in the target task. Table 16 illustrates the necessity of reward shaping by comparing performance with and without it. Note that reward shaping is applied consistently to all baseline algorithms as well as TraIRL.

Table 16: Reward shaping experiments on Assistive Gym environment. The reward function is learned in the source tasks (Feeding task 1 & 2).

|  | Scratch Itch | |
| --- | --- | --- |
|  | without Reward Shaping | with Reward Shaping |
| I2L | $-19.55 \pm 3.90$ | $-10.07 \pm 8.02$ |
| TraIRL | $-20.13 \pm 5.11$ | $-3.82 \pm 3.33$ |

### D.8   ABLATION STUDY

In this section, we conduct an ablation study on three objective functions, $\mathcal{L}_{\text{VAE}}, \mathcal{L}_{\text{WGAN}}, \mathcal{L}_{\mathcal{F}}$, in the overall objective function, Eq. 6.

Table 17: Ablation Study

|  | Sources | | Target |
| --- | --- | --- | --- |
|  | Run (rear disabled) | Run (front disabled) | Run (no disability) |
| Without $\mathcal{L}_{\text{VAE}}$ | $\mathbf{4,099.25 \pm 38.7}$ | $\mathbf{4,131.87 \pm 47.8}$ | $1,083.63 \pm 167.0$ |
| Without $\mathcal{L}_{\text{WGAN}}$ | $-133.89 \pm 211.8$ | $-118.35 \pm 259.4$ | $-181.2 \pm 203.6$ |
| Without $\mathcal{L}_{\mathcal{F}}$ | $-148.1 \pm 264.8$ | $-137.72 \pm 213.7$ | $-189.0 \pm 183.5$ |
| TraIRL | $4,404.07 \pm 57.6$ | $4,359.35 \pm 99.2$ | $\mathbf{5,835.11 \pm 74.0}$ |

Table 17 presents an ablation study evaluating the contribution of the three key components in the TraIRL objective: the VAE loss ($\mathcal{L}_{\text{VAE}}$), the discriminator loss ($\mathcal{L}_{\text{WGAN}}$), and the reward loss ($\mathcal{L}_{\mathcal{F}}$). Removing $\mathcal{L}_{\text{VAE}}$ results in minimal impact on performance in the source tasks, but leads to a significant drop in the target task performance, suggesting that a shared abstracted state space is essential for transfer. In contrast, removing either $\mathcal{L}_{\text{WGAN}}$ or $\mathcal{L}_{\mathcal{F}}$ leads to a complete failure of learning across all tasks, with highly negative returns. This indicates the critical roles of adversarial

training for enforcing optimality alignment and of reward fitting for capturing the expert policy structure. The full TraIRL objective achieves strong performance across all tasks, particularly in the target domain, demonstrating that all three components are necessary for effective transfer learning.

### D.9 Violation of the Structural Alignment Assumption in Theorem 2

As stated in Theorem 2, a violation of the structural alignment assumption directly leads to a violation of Eq. 8. Intuitively, when this assumption is not satisfied, the abstract state density of the target task is shifted relative to that of the source tasks. Such a shift prevents the learned abstract reward from generalizing correctly, resulting in degraded performance on the target task. The shifted target abstract state density can be mitigated through one-shot transfer learning, as described in Appendix D.6. We conducted an additional experiment to validate this claim, where source task 1: Ant (Leg 0 & 1 disabled), source task 2: Ant (Leg 0 & 3 disabled), and target task: Ant (Leg 1 & 2 & 3 disabled)

Table 18: Experiment on violation of the structural alignment assumption.

|         | Source 1          | Source 2          | Target             | One-shot transfer learning |
|---------|-------------------|-------------------|--------------------|----------------------------|
| AIRL-ME | $2399.66 \pm 89.1$ | $2231.09 \pm 87.6$ | $169.00 \pm 99.4$  | –                          |
| RIME    | $2471.90 \pm 92.4$ | $2308.14 \pm 81.5$ | $130.61 \pm 108.3$ | –                          |
| I2L     | $2853.80 \pm 73.8$ | $2786.90 \pm 79.4$ | $149.52 \pm 172.6$ | –                          |
| TraIRL  | $2815.93 \pm 63.2$ | $2936.52 \pm 95.5$ | $152.17 \pm 153.1$ | $1396.57 \pm 95.0$         |
| Expert  | $3391.57 \pm 279.1$ | $3303.99 \pm 341.0$ | $1655.42 \pm 256.2$ | $1655.42 \pm 256.2$        |

In this setting, none of the source tasks contains optimality information about Leg 0, which becomes the only functional leg in the target task. This causes the abstract state distribution in the target to shift significantly from those in the source tasks, violating the structural alignment assumption. This result provides direct empirical support for Theorem 2: when the structural alignment assumption is violated, the abstract state density shifts, leading to poor generalization. After applying one-shot transfer learning, the issue of shifted abstract state density is mitigated, and the performance improves substantially.

### D.10 Comprehensive Experiments

In this section, we present additional experiments that incorporate a larger set of source and target tasks.

Table 19: Extended experiments on **source** tasks in the **Ant** domain (part 1).

|         | Leg 0,1 disabled       | Leg 0,2 disabled       | Leg 0,3 disabled       | Leg 1,2 disabled       | Leg 1,3 disabled       |
|---------|------------------------|------------------------|------------------------|------------------------|------------------------|
| AIRL-ME | $2,420.31 \pm 75.2$    | $2,361.22 \pm 82.7$    | $2,231.09 \pm 67.6$    | $2,389.65 \pm 52.0$    | $2,140.84 \pm 75.3$    |
| RIME    | $2,691.04 \pm 55.6$    | $2,641.37 \pm 91.2$    | $2,708.14 \pm 81.5$    | $\mathbf{2,901.67 \pm 49.9}$ | $2,590.71 \pm 61.9$    |
| I2L     | $2,762.88 \pm 64.1$    | $\mathbf{2,801.55 \pm 69.8}$ | $2,786.90 \pm 79.4$    | $2,831.28 \pm 36.4$    | $2,620.47 \pm 84.5$    |
| TraIRL  | $\mathbf{2,962.84 \pm 49.5}$ | $2,774.88 \pm 68.9$    | $\mathbf{2,989.54 \pm 53.0}$ | $2,844.94 \pm 17.2$    | $\mathbf{2,733.25 \pm 56.9}$ |
| Expert  | $3,127.77 \pm 172.4$   | $3,590.57 \pm 158.2$   | $3,303.99 \pm 341.0$   | $3,312.12 \pm 304.3$   | $3,369.05 \pm 216.8$   |

Table 20: Extended experiments on **source** tasks in the **Ant** domain (part 2).

|         | Leg 2,3 disabled       | Leg 0,1,2 disabled     | Leg 0,1,3 disabled     | Leg 0,2,3 disabled     | Leg 1,2,3 disabled     |
|---------|------------------------|------------------------|------------------------|------------------------|------------------------|
| AIRL-ME | $2,302.44 \pm 78.1$    | $1,182.06 \pm 50.3$    | $1,160.92 \pm 48.6$    | $1,121.37 \pm 46.9$    | $1,131.58 \pm 52.1$    |
| RIME    | $2,660.08 \pm 79.6$    | $1,301.77 \pm 52.5$    | $\mathbf{1,483.49 \pm 36.8}$ | $1,241.62 \pm 50.7$    | $1,252.33 \pm 44.2$    |
| I2L     | $2,721.63 \pm 78.2$    | $1,381.94 \pm 40.1$    | $1,362.15 \pm 35.7$    | $\mathbf{1,361.08 \pm 38.9}$ | $1,332.76 \pm 45.0$    |
| TraIRL  | $\mathbf{2,802.07 \pm 42.4}$ | $\mathbf{1,454.11 \pm 29.6}$ | $1,418.54 \pm 18.3$    | $1,359.91 \pm 23.1$    | $\mathbf{1,361.90 \pm 50.6}$ |
| Expert  | $3,068.32 \pm 209.7$   | $1,716.93 \pm 227.0$   | $1,680.04 \pm 199.8$   | $1,652.81 \pm 161.2$   | $1,655.42 \pm 256.2$   |

Tables 19–21 present an extended experiment in the Ant domain, which includes 10 source tasks and 5 target tasks. In the source tasks, I2L and RIME occasionally achieve slightly higher returns than TraIRL, reflecting their ability to fit specific task instances. However, in the target tasks, TraIRL clearly outperforms all baselines, demonstrating that the transferable abstract state space

Table 21: Extended experiments on **target** tasks in the **Ant** domain.

|  | Leg 0 disabled | Leg 1 disabled | Leg 2 disabled | Leg 3 disabled | No leg disabled |
|---|---|---|---|---|---|
| AIRL-ME | $2,412.67 \pm 83.1$ | $2,395.42 \pm 102.7$ | $2,368.19 \pm 59.4$ | $2,401.55 \pm 97.2$ | $2,680.14 \pm 91.5$ |
| RIME | $2,675.34 \pm 72.8$ | $2,659.48 \pm 81.3$ | $2,621.77 \pm 69.5$ | $2,670.91 \pm 86.4$ | $2,945.63 \pm 68.2$ |
| I2L | $2,752.18 \pm 64.7$ | $2,736.02 \pm 65.0$ | $2,701.36 \pm 71.9$ | $2,749.85 \pm 79.6$ | $3,028.77 \pm 83.4$ |
| TraIRL | $\mathbf{3,241.03 \pm 95.0}$ | $\mathbf{3,237.62 \pm 64.3}$ | $\mathbf{3,217.80 \pm 80.6}$ | $\mathbf{3,276.18 \pm 75.7}$ | $\mathbf{3,546.96 \pm 161.1}$ |
| Expert | $3,619.15 \pm 139.7$ | $3,550.75 \pm 141.5$ | $3,440.70 \pm 123.7$ | $3,584.58 \pm 133.1$ | $4,091.59 \pm 159.2$ |

enables stronger generalization and more reliable reward transfer. This highlights the key advantage of TraIRL: while other methods may match or surpass performance on the sources, only TraIRL maintains superior performance when adapting to unseen target tasks.

Another extended experiment is conducted in the Assistive Gym environment[3], where source 1 (Feeding Sawyer with a static patient), source 2 (Feeding Sawyer with a tremor patient), source 3 (Feeding Sawyer with a static patient and a disabled wrist joint), source 4 (Feeding Sawyer with a tremor patient and a disabled wrist joint), source 5 (Scratch Itch Sawyer with a static patient), source 6 (Scratch Itch Sawyer with a tremor patient), source 7 (Scratch Itch Sawyer with a static patient and a disabled wrist joint), and source 8 (Scratch Itch Sawyer with a tremor patient and a disabled wrist joint); target 1 (Drinking Sawyer with a static patient), target 2 (Drinking Sawyer with a tremor patient), target 3 (Drinking Sawyer with a static patient with a static patient and a disabled elbow joint), and target 4 (Drinking Sawyer with a tremor patient and a disabled elbow joint).

Table 22: Extended experiments on **source** tasks in the Assistive Gym (part 1).

|  | Source 1 | Source 2 | Source 3 | Source 4 | Source 5 |
|---|---|---|---|---|---|
| AIRL-ME | $3.8 \pm 2.7$ | $4.5 \pm 6.9$ | $1.1 \pm 0.4$ | $4.0 \pm 2.3$ | $-5.6 \pm 2.1$ |
| RIME | $7.1 \pm 7.1$ | $7.8 \pm 6.0$ | $5.9 \pm 3.3$ | $7.2 \pm 1.1$ | $-2.9 \pm 4.8$ |
| I2L | $\mathbf{10.9 \pm 3.8}$ | $8.5 \pm 3.1$ | $\mathbf{10.0 \pm 5.5}$ | $8.7 \pm 6.2$ | $-2.5 \pm 0.7$ |
| TraIRL | $9.3 \pm 5.1$ | $\mathbf{11.1 \pm 4.2}$ | $8.7 \pm 6.1$ | $\mathbf{9.0 \pm 6.1}$ | $\mathbf{-2.0 \pm 1.6}$ |
| Expert | $11.2 \pm 5.3$ | $12.7 \pm 4.2$ | $9.54 \pm 3.1$ | $10.2 \pm 4.0$ | $-1.1 \pm 5.7$ |

Table 23: Extended experiments on **source** tasks in the Assistive Gym (part 2).

|  | Source 6 | Source 7 | Source 8 |
|---|---|---|---|
| AIRL-ME | $-6.9 \pm 2.8$ | $-6.4 \pm 1.9$ | $-6.7 \pm 1.0$ |
| RIME | $-3.9 \pm 3.7$ | $-3.4 \pm 4.8$ | $-3.6 \pm 5.9$ |
| I2L | $-3.1 \pm 0.6$ | $\mathbf{-2.9 \pm 0.7}$ | $-3.2 \pm 1.6$ |
| TraIRL | $\mathbf{-2.7 \pm 2.5}$ | $-3.0 \pm 2.6$ | $\mathbf{-2.8 \pm 2.5}$ |
| Expert | $-2.3 \pm 4.8$ | $-3.1 \pm 4.9$ | $-2.9 \pm 3.7$ |

Table 24: Extended experiments on **target** tasks in the Assistive Gym.

|  | Target 1 | Target 2 | Target 3 | Target 4 |
|---|---|---|---|---|
| AIRL-ME | $12.4 \pm 4.8$ | $11.2 \pm 2.5$ | $10.9 \pm 6.7$ | $11.3 \pm 5.6$ |
| RIME | $18.9 \pm 8.2$ | $17.5 \pm 10.0$ | $16.8 \pm 8.1$ | $17.2 \pm 7.2$ |
| I2L | $20.7 \pm 3.5$ | $24.4 \pm 5.2$ | $22.8 \pm 4.4$ | $20.1 \pm 3.3$ |
| TraIRL | $\mathbf{29.8 \pm 6.1}$ | $\mathbf{27.2 \pm 4.0}$ | $\mathbf{26.7 \pm 6.2}$ | $\mathbf{25.0 \pm 6.1}$ |
| Expert | $35.7 \pm 11.7$ | $30.5 \pm 9.4$ | $28.2 \pm 11.0$ | $25.7 \pm 10.2$ |

Tables 22–24 present extended experiments in the Assistive Gym domain, which include 8 source tasks and 4 target tasks. In the source tasks, I2L and RIME occasionally surpass TraIRL on certain instances, reflecting their strength in fitting task-specific structures. However, in the target tasks, TraIRL consistently outperforms all baselines, achieving results that are much closer to expert

---

[3]https://github.com/Healthcare-Robotics/assistive-gym/wiki

performance. This demonstrates that the transferable abstract state space learned by TraIRL enables stronger generalization and more reliable reward transfer. The results highlight the key advantage of TraIRL: while baselines can sometimes match or exceed performance in the sources, only TraIRL sustains superior generalization when transferring to unseen target tasks.

As the number of source tasks increases, the reward learned by TraIRL becomes more transferable because the abstraction is trained on a richer and more diverse set of trajectories. This diversity allows the abstract state space to capture higher-level features that are invariant across a broader range of variations, reducing the chance of overfitting to any single task. Consequently, the learned reward generalizes more effectively to unseen targets, as it encodes task-independent structure rather than idiosyncratic details. In practice, incorporating more source tasks also improves robustness, since the abstraction must reconcile multiple dynamics and objectives, leading to a more stable and transferable reward representation.

## E COMPUTING RESOURCES

All experiments were conducted on a desktop machine running Ubuntu 20.04, equipped with an Intel Core i7-10700K CPU, 32 GB of RAM, an NVIDIA RTX 3070 GPU, and CUDA 12.6.

## F VISUALIZATION OF ABSTRACTED STATE SPACE

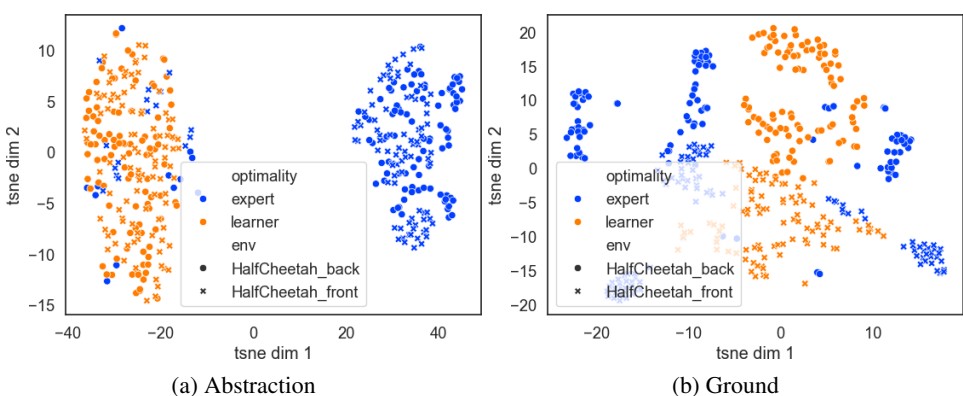

(a) Abstraction  (b) Ground

Figure 9: Visualization of **Half Cheetah** domain by t-SNE.

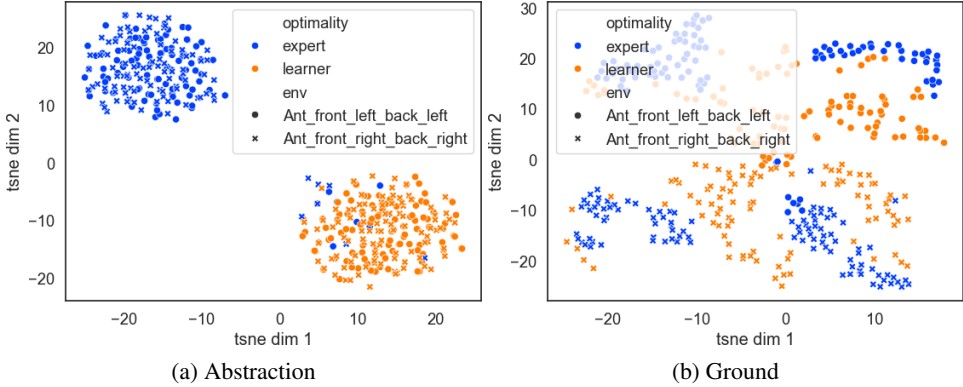

(a) Abstraction  (b) Ground

Figure 10: Visualization of **Ant** domain by t-SNE.

Fig. 9 presents t-SNE visualizations comparing the distributions of expert and learner states across two Half Cheetah tasks in both the ground and abstracted state spaces. In Fig. 9a, the expert and learner trajectories form well-separated clusters, orange for the learner and blue for the expert, while the trajectories from the two source tasks, Half Cheetah with rear legs disabled (circle) and with

front legs disabled (cross), appear largely overlapping in the abstract state space. The overlapping of trajectories across different tasks and the clear separation between the expert and the learner demonstrate that the abstract state space captures task-invariant features while preserving optimality information useful for reward learning. In Fig. 9b, both the expert and learner trajectories, as well as the two source tasks, form clearly separated clusters. This indicates the presence of task-specific features in the ground state space, which hinders the generalization of learned rewards from source tasks to the target task. The same conclusion can be made in the Ant domain in Fig. 10.

## G  LIMITATIONS

While TraIRL demonstrates strong generalization across related tasks, several limitations remain. First, the method assumes that source and target tasks share sufficient structural similarity. When this assumption is violated, such as in cases with large differences in state distributions or dynamics, the transferability of the learned reward may degrade. Balancing the training loss between source tasks also poses challenges, especially when their difficulty or distribution differs significantly. An imbalance can cause the abstract representation to overfit to one source task, reducing its effectiveness in the target.

Second, although TraIRL enables zero-shot transfer within the same task family, transferring to structurally different domains (e.g., from Ant to Half Cheetah) requires additional adaptation. This often involves learning mappings between ground states or introducing domain-specific constraints to mitigate distribution shifts.

Third, learning a reward function based on a common abstract representation may in some cases hurt performance. If the abstraction suppresses task-specific features that are critical for solving the target task, the resulting reward may fail to induce effective policies. As we showed in Sec 5.2, a goal-specific reward should be added on top of the abstract reward.

Finally, TraIRL introduces multiple interacting components—encoder, discriminator, and reward function—each with their own hyperparameters. Tuning these across tasks can be non-trivial. Overly complex discriminators or reward models may overfit to source tasks, harming generalization. Moreover, performance is sensitive to the balance among its three objectives: abstraction quality (VAE), optimality separation (discriminator), and reward alignment (reward function). Achieving this balance remains a key challenge for stable and effective training.

