# OpenReview forum: "Inversely Learning Transferable Rewards via Abstracted States"
_ICLR.cc/2026/Conference — ICLR 2026 Conference Withdrawn Submission_

### Official Review · Reviewer_1yzq · 2025-10-27

**Soundness:** 2
**Presentation:** 2
**Contribution:** 3
**Rating:** 2
**Confidence:** 4

**Summary:**

The authors propose a method for transfer inverse RL where they can transfer rewards to unseen but similar environments.  They use a multi-task VAE, with shared encoder and task-specific decoders, to learn a shared, abstract state space between all training tasks, and a WGAN discriminator to distinguish between expert and policy trajectories on the abstract state space.  This discriminator can then be transferred directly to a new task without new demonstrations.  They demonstrate this on MuJoCo-Gym locomotion tasks with different disabled limbs and on an Assistive Gym task.  They also provide theoretical results that link reward transferability to the abstract state densities between tasks.

**Strengths:**

1. The authors present a strong idea of learning abstract state representations for a shared reward function.
2. They present promising empirical evidence.  The reward transfer from Ant to HalfCheetah is especially surprising.

**Weaknesses:**

1. It's unclear how the multi-task VAE actually aligns the state representations between different tasks.  Even with the discriminative objective, it's entirely possible that the encoder does not well align semantically equivalent states.  There's some evidence that the learned embeddings do align based on the t-SNE plots but this may not be the case in higher dimensions.

2. The definition of the problem setting and what tasks can transfer rewards between are imprecise.  The locomotion experiments all deal with transferring the same locomotion tasks between characters with different active limbs or embodiments.  The Human Assistive gym tasks do transfer rewards between feeding and itching tasks but seem to require an extra task reward so it's unclear what is being transferred here.

3. Overall the experiments lack breadth.  Does the method generalize to more harder locomotion or manipulation tasks when the reward functions are more complex?  Can it handle other types of unseen environments (environment dynamics differences, obstacles, different reward functions).

**Questions:**

1. How is your method able to generalize to HalfCheetah from training on Ant tasks?
2. Is the encoder and reward function frozen for the target task?

---

> ### Author Response · Authors · 2025-11-19
> **Rebuttal**
>
> We thank the reviewer for the detailed review.
>
> ---
>
> 1. **How is your method able to generalize to HalfCheetah from training on Ant tasks?**
>
>     A semantic analysis of the learned abstract state (Appendix D.1) provides evidence for how our method generalizes from Ant to HalfCheetah. Since the reward function is reused without modification when transferring to HalfCheetah, successful generalization hinges on whether the encoder can produce an abstract state that remains aligned across both morphologies. Our analysis shows that, in the Ant environment, the abstract state is most sensitive to the x-direction velocity, precisely the factor that dominates optimal behavior in HalfCheetah as well. This consistency indicates that the encoder captures transferable motion dynamics, enabling the reward function learned on Ant to remain valid when applied to HalfCheetah. The details of transfer learning for Ant-to-HalfCheetah are represented in Appendix D.6.
>
> 2. **Is the encoder and reward function frozen for the target task?**
>     The reward function is kept frozen for all target tasks. The encoder, however, is frozen only when the transfer occurs within the same environment family (e.g., all source and target tasks are variants of Ant or all are variants of HalfCheetah). In the cross-morphology transfer from Ant to HalfCheetah, we train a new encoder because the state dimensionality differs between the two robots, making it impossible to reuse the original encoder directly.

---

> > ### Author Response · Authors · 2025-11-25
> >
> > We hope our reply has addressed your questions. We welcome any further discussion or clarification that may assist in your evaluation, and we remain fully open to addressing additional questions or concerns.
> >
> > Thank you for your time and consideration.

---

> > > ### Comment · Reviewer_1yzq · 2025-11-25
> > >
> > > Thank you for the responses to my questions.  My main concerns are still those stated under the Weaknesses section so I will be maintaining my score.

---

### Official Review · Reviewer_pTgQ · 2025-10-27

**Soundness:** 3
**Presentation:** 3
**Contribution:** 3
**Rating:** 6
**Confidence:** 4

**Summary:**

The paper introduces TraIRL, a method for learning transferable reward functions in inverse RL. The core idea is to learn a reward function over an abstract state space that is invariant to the dynamics of different source tasks. The method uses a multi-head VAE to learn this shared abstract representation from expert demonstrations. A Wasserstein GAN objective is then used to structure this abstract space by discriminating between expert and learner trajectories, guiding the learning of a reward function that captures task-agnostic intent. The learned abstract reward is then transferred to a novel target task to train a policy without requiring target-domain demonstrations.

**Strengths:**

*   **Principled Approach to Disentanglement:** The paper proposes a well-motivated method to disentangle a task's core reward from its specific dynamics by learning a reward function in an abstract state space. This is a significant conceptual strength.
*   **Strong Empirical Performance:** The method demonstrates superior performance over strong baselines in transferring rewards across tasks with different dynamics within the same domain (e.g., MuJoCo Ant with different disabled legs).
*   **Theoretical Grounding:** The paper provides a formal analysis (Theorem 2) that delineates the conditions under which the learned reward is transferable, linking performance to the structural properties of the abstract space.

**Weaknesses:**

*   **Unverifiable Theoretical Assumptions:** The main theoretical result, Theorem 2, hinges on the "structural alignment" assumption that optimal policies in source and target tasks are close in the abstract space. The paper provides no mechanism to verify this assumption for a new target task, rendering the guarantee non-constructive. The theory explains when transfer works, but provides no guidance on how to ensure it.
*   **Incomplete Reward Transfer:** In the AssistiveGym experiments, the learned abstract reward is insufficient to solve the target task and requires supplementation with an explicit, goal-specific reward shaping term. This suggests the method learns a useful behavioral prior (e.g., how to move smoothly) rather than a complete, transferable reward function that specifies the goal.
*   **High Complexity and Sensitivity:** The framework combines a VAE, a WGAN-GP, and a reward network, resulting in a complex system with many interacting components and hyperparameters. This complexity could pose a significant barrier to reproducibility and practical use.
*   **Misleading Cross-Domain Results:** The impressive Ant-to-HalfCheetah result in Table 2 is from a "one-shot" setting that requires a target expert trajectory and a different training objective (cycle loss), as detailed only in Appendix D.6. Presenting this in the main paper without context overstates the method's zero-shot capability.

**Questions:**

1.  Why was the Ant-to-HalfCheetah result in Table 2 presented without clarifying its one-shot nature in the main text? This seems to overstate the method's zero-shot transfer capabilities. Would you consider adding the zero-shot result to the table for a more complete and transparent comparison?
2.  The need for additional reward shaping in AssistiveGym suggests the learned "reward" is more of a behavioral prior. How does your method's performance (with shaping) compare to a simpler baseline that uses the same goal-based shaping but replaces the learned reward with a simple, handcrafted shaping term (e.g., action magnitude penalty)? This would help isolate the value of the learned abstract component.
3.  Regarding the structural alignment assumption: Is there any metric that can be computed from the learned VAE and source data to estimate the potential for transfer to a new target domain *before* committing to a full RL training run? Without this, the applicability of the method seems to rely on trial and error.

---

> ### Author Response · Authors · 2025-11-19
> **Rebuttal**
>
> We thank the reviewer for the detailed review.
>
> ---
>
> 1. **Why was the Ant-to-HalfCheetah result in Table 2 presented without clarifying its one-shot nature in the main text? This seems to overstate the method's zero-shot transfer capabilities. Would you consider adding the zero-shot result to the table for a more complete and transparent comparison?**
>
>     We have edited Table 2 to clarify the one-shot setup. We have mentioned that the details of the one-shot / zero-shot learning can be found in Appendix D.6 in line 380.
>
> 2. **The need for additional reward shaping in AssistiveGym suggests the learned "reward" is more of a behavioral prior. How does your method's performance (with shaping) compare to a simpler baseline that uses the same goal-based shaping but replaces the learned reward with a simple, handcrafted shaping term (e.g., action magnitude penalty)? This would help isolate the value of the learned abstract component.**
>
>     A comparison with a handcrafted shaping reward is not appropriate because it assumes that we already know the target task’s objective. In the IRL setup, we do not have such prior knowledge. If the target objective were known, we could simply handcraft a reward function for that task, and IRL would not be needed. The only information available to the algorithm is the expert demonstrations from the source tasks. TraIRL learns a behavioral prior that is shared across the source tasks. This prior is encoded in the abstract state representation and captures the common structure of expert behavior. In AssistiveGym, the target tasks have different goals compared to the source tasks, and the learned reward alone does not provide sufficient guidance for these new objectives. Therefore, we add reward shaping as an extra term. The reward shaping reflects the difference in goals across tasks, but it does not replace the learned reward. The shaping term alone is too sparse to train a policy, and it is only effective when combined with the behavioral prior that TraIRL has already learned from the source tasks.
>
> 3. **Regarding the structural alignment assumption: Is there any metric that can be computed from the learned VAE and source data to estimate the potential for transfer to a new target domain before committing to a full RL training run? Without this, the applicability of the method seems to rely on trial and error.**
>
>     For the current version of TraIRL, we do not provide a metric that can estimate transfer success before running the full RL training. However, the framework can be extended with an auxiliary component that evaluates how familiar the target-domain states are to the learned abstraction. One possible approach is inspired by Random Network Distillation (RND). In RND, a predictor network learns to approximate the output of a fixed random network. The prediction error between the predictor and the random network is used as a measure of confidence. A small prediction error indicates that the state lies in a region that is well covered by the source tasks, while a large error indicates an unfamiliar or out-of-distribution region. This RND prediction error can serve as a metric to estimate whether the learned abstraction, and therefore the learned reward, is likely to transfer successfully to a new target domain. If the RND error is consistently large on target-domain states, the method can flag the transfer as unlikely to succeed before committing to a full RL training run.

---

> > ### Author Response · Authors · 2025-11-25
> >
> > We hope our reply has addressed your questions. We welcome any further discussion or clarification that may assist in your evaluation, and we remain fully open to addressing additional questions or concerns.
> >
> > Thank you for your time and consideration.

---

### Official Review · Reviewer_9uju · 2025-10-31

**Soundness:** 3
**Presentation:** 3
**Contribution:** 2
**Rating:** 4
**Confidence:** 4

**Summary:**

This paper addresses the problem of reward transfer between environments in inverse reinforcement learning (IRL). To learn transferable rewards, the paper introduces the TraIRL method, which first learns a state abstraction and then learns a reward using a standard IRL framework within this abstracted state space. Experiments across two MuJoCo domains demonstrate that TraIRL can learn rewards that transfer effectively to unseen environments.

**Strengths:**

1. The paper investigates the important problem of reward transfer between related tasks in IRL. It is crucial that rewards learned through IRL generalize to unseen settings.
2. The paper formally studies the problem of reward transfer in Section 4.5.
3. The comparisons in MuJoCo Gym and Assistive Gym show that TraIRL outperforms baselines in generalization to new target tasks. The analysis in Section 5.2.1 also confirms that TraIRL learns a meaningful state abstraction.

**Weaknesses:**

1. TraIRL lacks novelty compared to prior IRL algorithms. The method first learns a state encoding and then performs standard IRL on top of this learned representation. Simply learning a state encoding before applying IRL offers limited advancement over prior work.
2. The experiments are limited to domains that are already well suited for learning an easily transferable state encoding. Both domains use the ground-truth state. In MuJoCo Gym, the state encoder only needs to ignore the joint information and focus on the torso, as described in Section 1. Thus, the state encoder can merely filter out irrelevant observations. The paper does not compare to the simple baseline of manually removing these clearly non-transferable features in reward learning. A more meaningful evaluation would involve complex observation spaces where learning a transferable state representation for rewards is challenging, such as image-based observations.
3. The paper lacks sufficient empirical evaluation. Results are presented for only three settings. Additional evaluations are needed to fully assess TraIRL’s performance.
4. The paper does not report the performance of f-IRL, which TraIRL uses as its underlying IRL algorithm.
5. Reporting performance based on only ten episodes per task is insufficient. Since the experiments are conducted in fast simulated environments, many more episodes should be used for evaluation.
6. The paper does not examine how performance changes as the number of expert trajectories varies. Does TraIRL maintain strong performance relative to baselines as the number of trajectories increases? Do baselines also learn good state abstractions when provided with more expert demonstrations? Is TraIRL still able to learn state abstractions with fewer demonstrations?

**Questions:**

1. How does TraIRL compare to simply removing joint information from the state and retaining only the torso information for reward learning in MuJoCo Gym?
2. What does “ground state density” refer to in line 391?
3. Why are the baseline results omitted for Half Cheetah in Table 2?
4. What is the performance impact of updating the encoder with the reward, as mentioned in line 243?
5. How does TraIRL’s performance compare to regular f-IRL?
6. Do the baselines also jointly train the reward across multiple tasks as TraIRL does?

---

> ### Author Response · Authors · 2025-11-20
> **Rebuttal**
>
> We thank the reviewer for the detailed review.
>
> ---
>
> 1. **How does TraIRL compare to simply removing joint information from the state and retaining only the torso information for reward learning in MuJoCo Gym?**
>
>     We would like to highlight an important point: restricting the reward learning input to only the torso-related features implicitly introduces prior knowledge into the model, as it filters out information that may be relevant. This contradicts our problem setup, where no prior knowledge regarding the reward function should be assumed before performing the experiments. In parallel, we have conducted experiments where only the torso information is reserved.
>
>     Table 1. Half Cheetah
>     || Run (rear disabled, source env) | Run (front disabled, source env) | Run (no disability, target env) |
>     |---|---|---|---|
>     | AIRL (Torso Only) | 4138.77 +- 105.5 | 4110.10 +- 173.9 | 5302.66 +- 241.3 |
>     | TraIRL (Torso Only) | 4102.83 +- 143.4 | 4007.56 +- 191.0 | 5279.01 +- 218.2 |
>     | TraIRL | 4404.07 +- 57.6 | 4359.35 +- 99.2 | 5853.11 +- 74.0 |
>
>     Table 2. Ant
>     || Leg 1,2 disabled (source env) | Leg 0,3 disabled (source env) | Leg 1,3 disabled (target env) | Leg 0,2 disabled (target env) | Half Cheetah (directly reuse trained Encoder & Reward, target env) | Half Cheetah (One-Shot, target env) |
>     |---|---|---|---|---|---|---|
>     | AIRL (Torso Only) | 2339.53 +- 101.4 | 2404.77 +-119.5  | 2254.49 +- 95.2 | 2195.78 +- 111.6 | 3024.10 +- 93.5 | 4838.98 +- 159.1 |
>     | TraIRL (Torso Only) | 2342.91 +- 128.7 | 2374.09 +- 135.5 | 2308.60 +- 110.3 | 2355.71 +- 157.0 | 2806.17 +- 173.2 |  4905.82 +-  167.3 |
>     | TraIRL | 2714.18 +- 35.9 | 2936.52 +- 95.5 | 2917.92 +- 79.3 | 3156.54 +- 63.1 | - | 5378.78 +- 61.7 |
>
>     If the observation includes only torso information, the reward function cannot penalize joints that move outside their normal range, which leads to unnecessarily high action costs.
>
> 2.  **What does “ground state density” refer to in line 391?**
>
>     State density refers to the state occupancy measure, i.e., the state marginal distribution as defined in line 120. The ground state density corresponds to the marginal distribution of the ground-truth states (rather than the abstract states).
>
> 3. **Why are the baseline results omitted for Half Cheetah in Table 2?**
>
>     The baseline results are omitted for HalfCheetah because the baseline model is pre-trained on Ant, and the two environments have incompatible state dimensionalities. Consequently, the Ant-trained baseline cannot be directly applied or evaluated in the HalfCheetah environment. In contrast, TraIRL separates the encoder from the reward function, which allows us to learn a new encoder for environments with different state dimensions, such as the Ant-to-HalfCheetah transfer, while still reusing the same reward function.
>
> 4. **What is the performance impact of updating the encoder with the reward, as mentioned in line 243?**
>
>     As mentioned in line 246, this is a distillation process. The reward distills the knowledge from the discriminator without changing the input, i.e., the abstract state generated from the encoder. The entire goal of updating the reward without the encoder is to make the reward robust to the abstract state.
>
> 5. **How does TraIRL’s performance compare to regular f-IRL?**
>
>     Table 3. Half Cheetah
>     || Run (rear disabled, source env) | Run (front disabled, source env) | Run (no disability, target env) |
>     |---|---|---|---|
>     | f-IRL |  3252.72 +- 67.9 | 3523.01 +- 91.1 | 3582.10 +- 94.3 |
>     | TraIRL | 4404.07 +- 57.6 | 4359.35 +- 99.2 | 5853.11 +- 74.0 |
>
>     Table 4. Ant
>     || Leg 1,2 disabled (source env) | Leg 0,3 disabled (source env) | Leg 1,3 disabled (target env) | Leg 0,2 disabled (target env)
>     |---|---|---|---|---|
>     | f-IRL |  2456.17 +- 85.0 | 1,146.39 +- 95.8 | 1598.24 +- 44.9 |  1652.89 +- 74.3 |
>     | TraIRL | 2714.18 +- 35.9 | 2936.52 +- 95.5 | 2917.92 +- 79.3 | 3156.54 +- 63.1 |
>
> 6. **Do the baselines also jointly train the reward across multiple tasks as TraIRL does?**
>
>     Yes. There is only one reward function, and the reward is jointly trained across multiple tasks.

---

> > ### Author Response · Authors · 2025-11-25
> >
> > We hope our reply has addressed your questions. We welcome any further discussion or clarification that may assist in your evaluation, and we remain fully open to addressing additional questions or concerns.
> >
> > Thank you for your time and consideration.

---

### Note · Authors · 2026-01-15

I have read and agree with the venue's withdrawal policy on behalf of myself and my co-authors.